# Extra-embryonic tissue spreading directs early embryo morphogenesis in killifish

Germán Reig[1,2], Mauricio Cerda[1,2], Néstor Sepúlveda[3], Daniela Flores[1,2], Victor Castañeda[1,2], Masazumi Tada[4], Steffen Härtel[1,2,5] & Miguel L. Concha[1,2,6]

The spreading of mesenchymal-like cell layers is critical for embryo morphogenesis and tissue repair, yet we know little of this process *in vivo*. Here we take advantage of unique developmental features of the non-conventional annual killifish embryo to study the principles underlying tissue spreading in a simple cellular environment, devoid of patterning signals and major morphogenetic cell movements. Using *in vivo* experimentation and physical modelling we reveal that the extra-embryonic epithelial enveloping cell layer, thought mainly to provide protection to the embryo, directs cell migration and the spreading of embryonic tissue during early development. This function relies on the ability of embryonic cells to couple their autonomous random motility to non-autonomous signals arising from the expansion of the extra-embryonic epithelium, mediated by cell membrane adhesion and tension. Thus, we present a mechanism of extra-embryonic control of embryo morphogenesis that couples the mechanical properties of adjacent tissues in the early killifish embryo.

[1] Anatomy and Developmental Biology Program, Institute of Biomedical Sciences, Faculty of Medicine, Universidad de Chile, PO Box 70031, Santiago, Chile. [2] Biomedical Neuroscience Institute, Independencia 1027, Santiago, Chile. [3] Department of Physics, Faculty of Physical and Mathematical Sciences, Universidad de Chile, PO Box 487-3, Santiago, Chile. [4] Department of Cell and Developmental Biology, University College London, Gower Street, London WC1E 6BT, UK. [5] National Center for Health Information Systems CENS, Independencia 1027, Santiago, Chile. [6] Center for Geroscience, Brain Health and Metabolism, Las Palmeras 3425, Ñuñoa, Santiago, Chile. Correspondence and requests for materials should be addressed to M.L.C. (email: mconcha@med.uchile.cl).

Most of our knowledge on the mechanisms controlling the spreading of mesenchymal-like tissues comes from *in vitro* studies where cells use extracellular matrix (ECM)-based substrates to migrate. *In vivo*, however, cells frequently use the surface of other cells to migrate yet the basic principles governing this process remain poorly understood. The inherent complexity of composite-tissue systems, especially during embryonic development, has challenged our ability to dissect the contribution of autonomous (that is, intrinsic cell/tissue properties) from non-autonomous (that is, physical/chemical environmental signals) mechanisms of tissue spreading. To circumvent this, we took advantage of unique developmental features of a non-conventional teleost embryo where undifferentiated mesenchymal-like embryonic cells spread as a collective at very low density and in a simple cellular environment, well before the onset of gastrulation and embryonic axis formation[1–3].

In annual killifish, the deep cell layer (DCL) of embryonic cells spreads from the animal pole to cover the entire surface of the egg. This event takes place in a space lacking organized ECM[4] and delimited by the squamous epithelial enveloping cell layer (EVL) and the yolk syncytial layer (YSL; Fig. 1)[2]. As in other teleost embryos, these two extra-embryonic tissues also undergo vegetal spreading and, together with the embryonic DCL, engulf the egg in a process known as epiboly[5–7]. The EVL, which lies above the DCL, spreads by an actomyosin-dependent mechanism generated at the circumferential margin that generates pulling forces and tension anisotropy within the epithelium[8,9]. In contrast, the mechanisms of DCL spreading are less understood although the current model based on zebrafish work proposes that this process relies on tissue-autonomous properties, where radial cell intercalation provides the driving force[10–13]. However, DCL spreading in annual killifish takes place in a context of low cellular density where intercellular space is available throughout the process (Fig. 1). Therefore, in contrast to zebrafish, radial cell intercalation is dispensable for DCL spreading in annual killifish and thus, other cellular mechanisms must operate to direct the epibolic spreading of this layer.

Here we reveal that the extra-embryonic EVL directs cell migration and the spreading of the embryonic tissue during epiboly. This function is mediated by cell adhesion and tension and relies on the ability of embryonic cells to couple their autonomous random motility to non-autonomous mechanical signals arising from the expansion of the EVL, which is used by embryonic cells as a substrate for migration. Thus, we present a previously undescribed function for extra-embryonic control of embryo morphogenesis that couples the mechanical properties of adjacent tissues in the early vertebrate embryo.

## Results

**Morphogenetic coupling of DCL and EVL spreading.** To start dissecting the mechanisms of DCL spreading we analysed the dynamics of this process and its relation to the expansion of adjacent extra-embryonic tissues, in particular, the EVL. For this, we tracked DCL movements and followed the shape changes of the EVL from late blastula to 60% epiboly using four-dimensional (4D)-confocal image data sets obtained from animal pole views (Fig. 2a, Supplementary Movie 1, Methods). During this period, the extra-embryonic EVL contained 50–60 polygonal-shape cells of variable size that progressively expanded their surface area (Fig. 2b) without events of cell division (Supplementary Movie 1). The embryonic DCL comprised 100–120 relatively small cells (Fig. 2b) that doubled their number after a round of asynchronous cell division (Supplementary Fig. 1a) and progressively increased the mean distance with their immediate neighbours (Fig. 2d; Supplementary Fig. 1b). Notably, the total area covered by the DCL increased as a linear function of the total EVL surface area ($r^2 = 0.9916$) (Fig. 2c; Supplementary Fig. 2c) and, in the

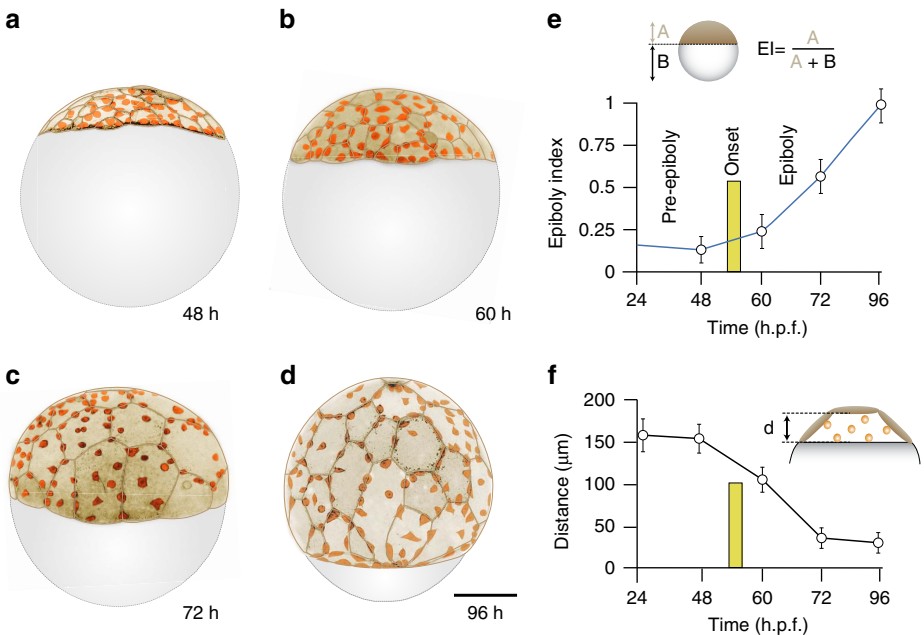

**Figure 1 | Stages of epiboly in the annual killifish *Austrolebias nigripinnis*.** (**a–d**) Lateral views of embryos expressing *lifeact-GFP* at late blastula (**a**) and during different stages of epiboly: 30–35% (**b**), 60–65% (**c**) and 85–90% (**d**). Images correspond to confocal microcopy z-stack maximum projections, with an inverted look-up table, and pseudocoloured in light brown for the EVL and orange for cells of the DCL. (**e**) Temporal changes in the epiboly index, defined by the position of the blastoderm margin along the animal–vegetal axis of the embryo (see formula at the top). (**f**) Temporal changes in the height of the blastocyst cavity, defined by the distance between the EVL basal surface and the inner surface of the YSL along the animal–vegetal axis. Numbers in (**a–d**) indicate hours post-fertilization. The vertical yellow line in **e** and **f** indicates the onset of epiboly, defined by the initial vegetal-ward movement of the EVL epithelial margin. Scale bar, 250 μm.

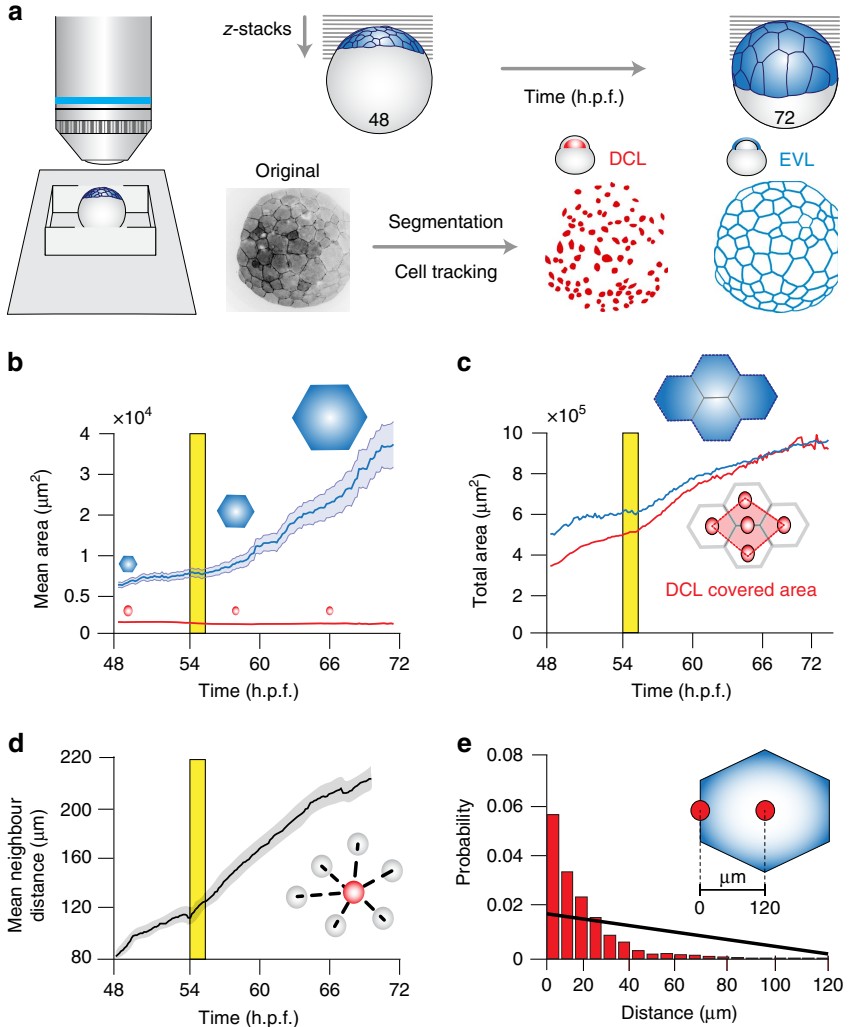

**Figure 2 | Dynamics of tissue spreading of the embryonic DCL and its relation to the surface expansion of the extra-embryonic EVL.** (**a**) Schematic diagram of *in vivo* imaging approaches. Embryos of *A. nigripinnis* expressing *lifeact-GFP* were imaged from the animal pole by 4D confocal microscopy between late blastula (48 h.p.f.) and 60% epiboly (72 h.p.f.), and cells of the DCL and EVL segmented and tracked over time. (**b**) Temporal changes in the mean surface area of EVL and DCL cells, with values expressed as means ± s.e.m. The blue polygons and red circles above the curves depict representative EVL and DCL cells drawn to scale, respectively. (**c**) Temporal changes in the total area covered by the EVL and the convex area covered by the underlying DCL cells, measured as indicated in the top and bottom insets, respectively. (**d**) Temporal changes in the mean distance to the six nearest neighbours within the DCL, with values expressed as means ± s.e.m. (**e**) Probability distribution of DCL cell position as a function of the distance to EVL cell borders, for the period between 30 and 60% epiboly (54–72 h.p.f.). The black straight line corresponds to the expected random distribution for a mean EVL cell surface radius of 120 µm, as indicated in the top right corner. In all panels, red and blue colours correspond to the DCL and EVL, respectively. The vertical yellow line in **b**–**d** indicates the onset of epiboly, defined by the initial vegetal-ward movement of the EVL epithelial margin. All panels of this figure were obtained after tracking and analysis of the entire set of EVL and DCL cells from Supplementary Movie 1. This analysis was replicated in a second embryo (Supplementary Fig. 2).

process, DCL cells distributed preferentially along the borders of EVL cells (Fig. 2e; Supplementary Fig. 3; Supplementary Movie 1). Such remarkable spatial configuration was maintained throughout epiboly despite the continuous expansion of the EVL cell surface and resulted in polygonal-shape cellular arrangements that mimicked the shape changes exhibited by EVL cells (Supplementary Fig. 3). Together, these results reveal that spreading of the embryonic DCL involves a process of cell dispersion that parallels closely to the epibolic expansion of the extra-embryonic EVL. This finding raises the questions of whether DCL cells interact with the EVL and if these interactions direct DCL spreading.

**Adhesive interactions between the DCL and EVL basal surface.** To address whether the DCL and EVL physically interact,

we performed high-resolution confocal imaging in embryos expressing membrane-tagged EGFP. We observed that when cells of the DCL contacted the EVL basal surface, they flattened at the contact side while remaining curved at the opposite contact-free side (Fig. 3b). We also noticed that when an EVL cell was induced to undergo apical extrusion (Fig. 4a), the surrounding DCL cells became progressively stretched and moved in synchrony with the contracting border of the extruding EVL cell (Fig. 4b–e; Supplementary Movie 2). Together, these findings suggest that the DCL and EVL establish adhesive contacts and that these contacts can transmit traction forces between the two cell layers. We then examined the function of E-cadherin/Cadherin-1 (E-cad), a cell adhesion protein known to participate in spreading of the DCL in zebrafish[11–13]. Cells of the DCL and EVL expressed *e-cad* throughout epiboly (Fig. 5a–c) and localized E-cad at the

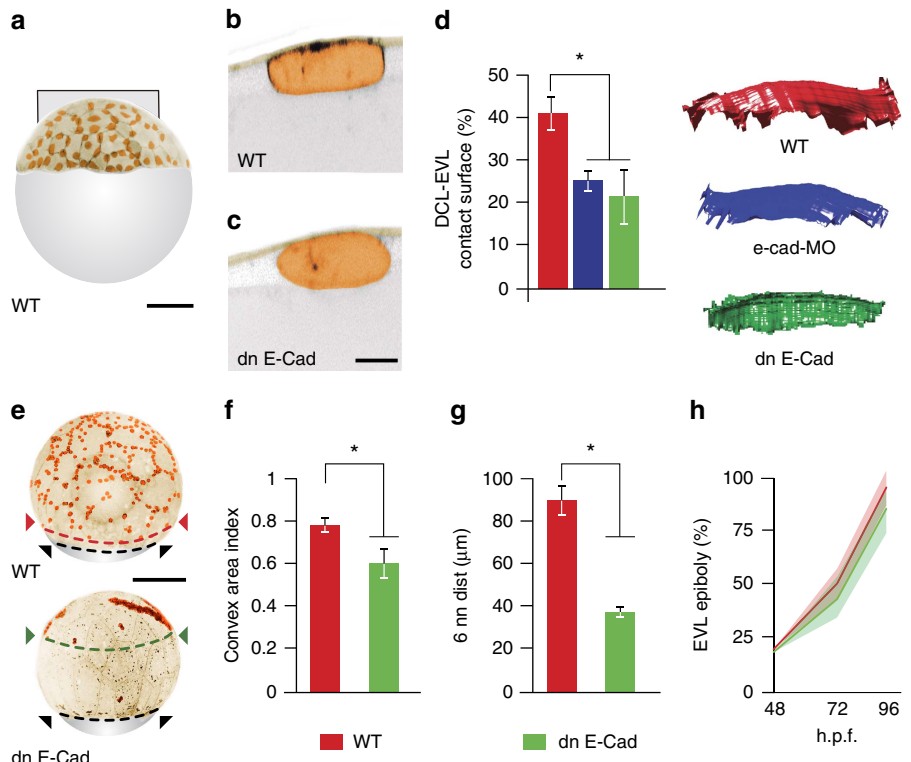

**Figure 3 | Adhesive interactions between the DCL and EVL during tissue spreading.** (**a**) Confocal microscopy z-stack maximum projection showing a lateral view of an *A. nigripinnis* embryo expressing *GAP43-EGFP*, with animal to the top, showing the distribution of DCL cells at 40% epiboly. (**b,c**) High-magnification confocal microscopy sections showing adhesive contacts established between the DCL and the EVL basal surface in the WT (**b**) and after functional abrogation of E-cad through the injection of dn E-cad (**c**). (**d**) Quantification of the DCL-EVL contact area (left), with corresponding examples of 3D volumetric projections (right), for the experimental conditions depicted in **b** and **c** (n = 6 cells per condition). (**e**) Confocal microscopy z-stack maximum projections showing lateral views of WT (top) and dn E-cad (bottom) embryos at 80% epiboly. Dashed lines indicate the advancement of the margin of the EVL (black) and DCL (red and green for WT and dn E-cad conditions, respectively). The look-up table of images in **a**–**c** and **e** has been inverted. EVL and DCL cells are pseudocoloured in light brown and orange, respectively. (**f,g**) Quantification of the total covered area (**f**) and mean distance to the six nearest neighbours (**g**) for the DCL in WT and dn E-cad embryos. (**h**) Temporal progression of the EVL margin at different stages of epiboly in WT and dn E-cad embryos. Values in plots (**f**–**h**) are expressed as means ± s.e.m. (n = 8 embryos per condition). *P < 0.001 (t-test). Scale bar, 30 μm (**b,c**); and 250 μm (**a,e**).

cell membrane, membrane protrusions and epithelial cell junctions (Fig. 5d,e). Functional abrogation of E-cad either through the injection of a specific morpholino antisense oligonucleotide (*e-cad*-MO; Supplementary Fig. 4) or by overexpression of dominant-negative E-cad (dn E-Cad)[14] (Supplementary Fig. 5) resulted in impairment of DCL-EVL adhesive contacts. DCL cells became spherical, did not flatten at DCL-EVL contacts and reduced the contact area with the EVL (Fig. 3c,d; Supplementary Fig. 5). Also, they were unable to follow the changes in position and shape exhibited by the EVL, compared to controls (Fig. 4d–f). Importantly, the vegetal spreading of the complete DCL was impaired by E-cad inactivation while the vegetal movement of the EVL margin remained unaffected (Fig. 3e–h). Thus, we conclude that the DCL establishes functional E-cad-mediated cell–cell adhesions with the extra-embryonic EVL basal surface which are required for spreading of the DCL during epiboly.

**DCL migration becomes directional at EVL cell borders.** As DCL cells adhere to the EVL, the process of DCL spreading could merely result from non-autonomous dragging by the extra-embryonic EVL as it expands during epiboly. Alternatively, autonomous cell migration could play an active role and either boost or oppose the dragging force exerted by the EVL. To dissect among these possibilities, we estimated the autonomous movement of the DCL (Fig. 6a,b and Methods). We found that DCL cells were not stably anchored to the EVL but moved with an autonomous random walk pattern ($D = 31.2 \, \mu m^2 \, s^{-1}$, $r^2 = 0.9757$). Coupling of autonomous cell migration to the expansion of the EVL basal surface thus increased the mean velocity, vegetal-ward directionality and mean square displacement of DCL cells (Fig. 6c–e). We also observed that although the overall movement of the DCL was random, individual cells exhibited transient cycles of directed motion towards EVL cell borders within a narrow preferential zone of ± 40 μm width along these regions (Fig. 6f,g). This migratory behaviour was based on F-actin-rich filopodial-like polarized protrusions (Fig. 6f and Supplementary Movie 3) and required the activity of Rac1. Indeed, DCL cells stopped moving at EVL cell borders and displayed randomly oriented thin membrane protrusions when overexpressing the dominant-negative Rac1-T17N (Fig. 6h; Supplementary Movie 4). Furthermore, DCL cells often presented actin brushes at transient contacts with EVL cell borders (Supplementary Fig. 6; Supplementary Movie 5) indicating their ability to sense these regions. Consistent with this idea, as a DCL cell approached the EVL cell border, it increased polarized protrusions and movement directionally (Fig. 7b–d; Supplementary Movie 6). After crossing the EVL cell border, the DCL cell re-polarized and moved back again towards the border

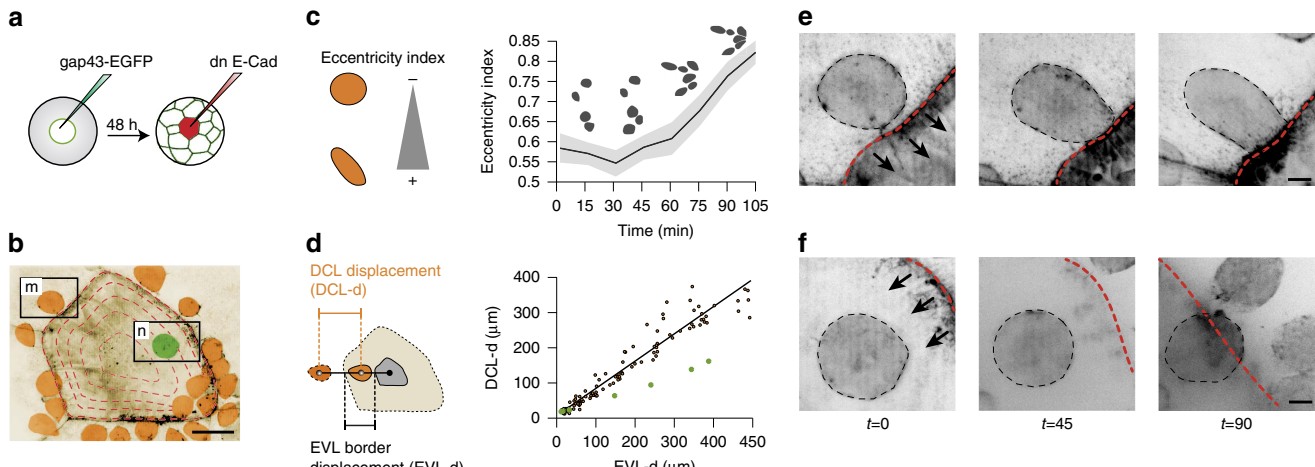

**Figure 4 | Traction forces exerted by the EVL on the DCL.** (**a**) Schematic diagram of the method used to induce EVL apical extrusion by overexpressing dn E-cad in a single EVL cell (red) in an *A. nigripinnis* embryo expressing *GAP43-EGFP*. (**b**) Confocal microscopy z-stack maximum projection centred in a EVL cell manipulated as in **a** at the onset of apical extrusion (time = 0). During the following 105 min, the EVL cell progressively contracts and reduces its surface area (dashed orange lines). The image look-up table has been inverted, the EVL pseudocoloured in light brown and DCL cells coloured according to their position: in the periphery (orange) and under (green) the extruding EVL cell. (**c**) Quantification of the eccentricity index for the DCL cells shown in (**b**), revealing cell elongation during EVL cell extrusion. Values are expressed as means ± s.e.m. (n = 20 cells). Examples of DCL cells at 4 time points are given above the curve. (**d**) Correlation between the displacement of the DCL centre of mass and the EVL cell border from time 0 to 105 for all DCL cells labelled in orange in **b** (n = 20 cells), showing the changes in DCL position with respect to the border contraction of the extruding EVL cell. Green dots indicate the changes in position for the green cell shown in **b**. (**e,f**) Time series of confocal microscopy z-stack maximum projections of two DCL cells selected from **b**, showing the changes in cell shape and position with respect to the border contraction of the extruding EVL cell. In **e**, the DCL is positioned in the periphery of the extruding EVL cell under a wild type EVL (with normal E-cad function). In **f**, the DCL is positioned under the extruding EVL cell (with impaired E-cad function). Arrows indicate the movement direction of the contracting border of the extruding EVL cell (dashed red line). Time is in minutes. Images and plots were extracted from Supplementary Movie 2. Scale bar, 10 μm (**e,f**) and 50 μm (**b**).

(Supplementary Movie 6). Together, these findings indicate that DCL cells show autonomous random motility when they are far from the EVL cell border and that they switch from random to directional migration when they approach this region. Thus, EVL cell borders provide short-range cues to attract DCL cell migration.

**Mechanical signals attract DCL cells to EVL cell borders.** The nature of signals involved in EVL cell border attraction could in principle be mechanical, biochemical or both. Although not excluding a possible function of biochemical cues, three lines of evidence suggest a primary role of mechanical signals in the process. First, we observed that the mechanical properties of the EVL basal surface (the substrate for DCL spreading) were distinct at EVL cell borders compared to non-border regions. Cortical tension, which determines the stiffness of the cell cortex and thus the resistance to cell-surface deformation[15], was increased at EVL cell borders. This was suggested by the enhanced distribution at EVL cell borders of phosphorylated myosin (Supplementary Fig. 7), which is a major determinant of cell contractility and thus cortical tension[15], and by the reduced deformation of the EVL basal surface at EVL cell borders compared to non-border regions (Fig. 7a). Second, we noticed a spatiotemporal matching between the shift in DCL migration from random to directional and a sharp decrease in EVL basal surface deformation (Fig. 7b,g). Strikingly, these changes were paralleled by a sharp increase in DCL-EVL contact surface (Fig. 7e,f), suggesting that as the DCL cell approaches the EVL cell border it increases the strength of adhesive interactions with the EVL basal surface. This change was not accompanied by a pre-configured gradient of E-cad across the EVL cell (Supplementary Fig. 8), which argues against an E-cad-dependent haptotactic mechanism and suggests that the increase in adhesive contacts might be a response to increased

substrate tension and/or stiffness, as previously reported for cells migrating *in vitro*[16,17]. We can not rule out, however, the existence of a gradient of another cell adhesion protein. Third, we observed that DCL cells adjusted their shape and migratory behaviour following local changes in tension within the EVL. For example, when EVL cells underwent rapid transient contractions of the cell cortex during events of failed cytokinesis (Supplementary Movie 7), when two EVL cells underwent fusion and the epithelial cell vertices retracted (Supplementary Fig. 9; Supplementary Movie 8), and when a single EVL cell experienced extrusion from the epithelium and tension built around the contracting actomyosin ring (Fig. 4b–e; Supplementary Movie 2). Altogether, these findings suggest that cortical tension and/or stiffness is enhanced at EVL cell borders and that this local property of the cellular substrate favours adhesive and tensile cell–substrate interactions that foster the migration of DCL cells.

To directly test whether tensile forces from the EVL play an instructive role in regulating the spatial distribution of DCL cells, we locally manipulated tension in the EVL by tuning actomyosin contractility and assessed the response of DCL cells. Remarkably, we observed that increasing tension in a single EVL cell by overexpressing RhoA[18] resulted in an increased density of DCL cells positioned under the modified EVL cell when compared to the surrounding regions (Fig. 8a–c). In the reverse experiment, the local decrease of EVL tension by overexpression of the N-terminal region of the myosin phosphatase target subunit 1 (N-ter-MYPT1), which reduces the level of phosphorylated myosin[19] and disrupts the formation of actin cables along EVL cell borders (Supplementary Fig. 10), led to reduced density of DCL cells under the modified EVL cell (Fig. 8b,c). Notably, local relaxation of EVL tension was transmitted across the plane and induced a cell non-autonomous disruption of DCL organization beyond the manipulated EVL cell (Fig. 8e), further supporting the

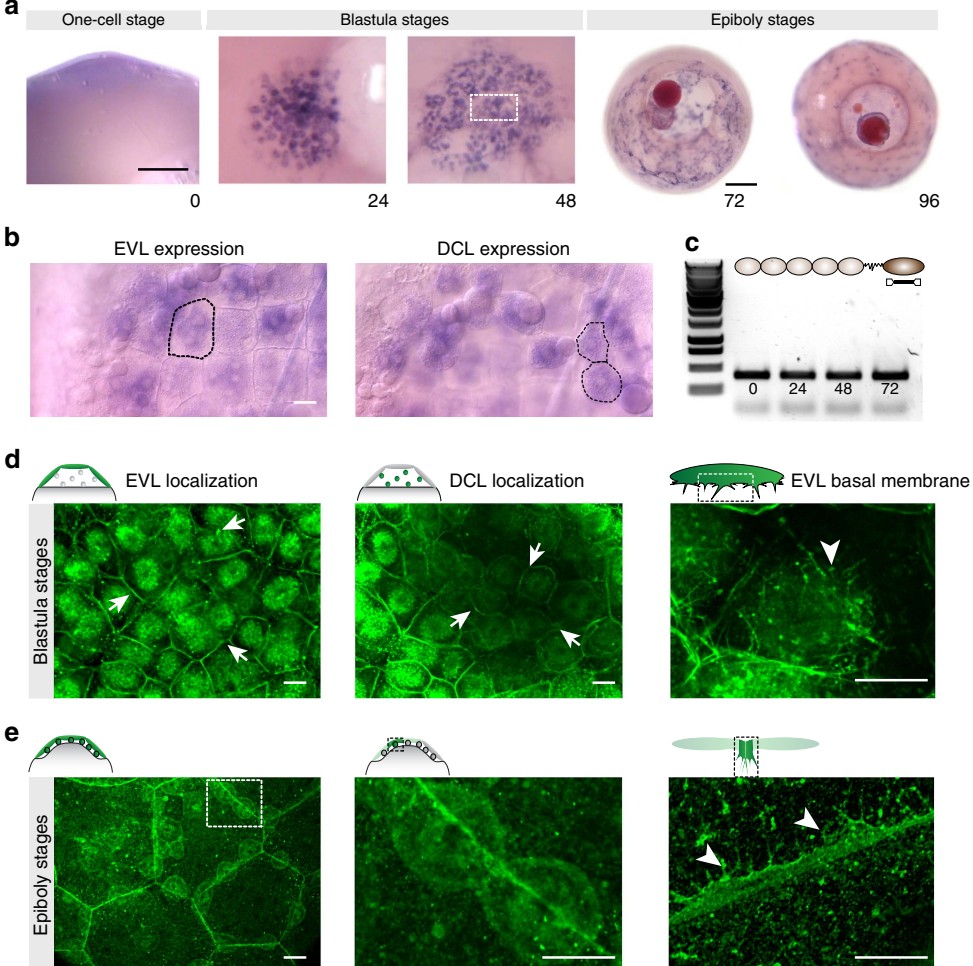

**Figure 5 | e-cad mRNA expression and E-cad protein localization during blastula and epiboly stages of annual killifish.** (**a**) Bright-field images of *A. nigripinnis* embryos after whole-mount *in situ* hybridization showing *e-cad* mRNA expression in violet from one-cell stage (0 h.p.f.) through blastula (24 and 48 h.p.f.) to epiboly (72 and 96 h.p.f.) stages. *e-cad* is expressed maternally. During blastula and epiboly stages, *e-cad* mRNA is detected at high levels in the EVL and DCL. All panels correspond to animal pole views with the exception of the 1-cell stage, which is a lateral view. (**b**) High-magnification views of the region depicted in **a** at 48 h.p.f. (dashed rectangle), showing the expression of *e-cad* in single cells of the EVL and DCL, outlined in left and right panels, respectively. (**c**) Semi-quantitative PCR showing the expression of a cDNA fragment of ± 300 bp coding for a region of the cytoplasmic domain of E-cad. (**d**) Confocal microcopy optical sections showing the localization of E-cad protein during blastula stages, after indirect immunofluorescence using a specific antibody against *A. nigripinnis* E-cad. E-cad protein concentrates at the cell membrane (arrows) and peri-nuclear regions of the EVL and DCL, with additional localization in filopodial protrusions from the DCL and EVL basal surface (arrowheads). (**e**) Confocal microcopy *z*-stack maximum projections showing the localization of E-cad protein during epiboly stages. E-cad becomes primarily localized at the cell membrane in the EVL and DCL, with prominent localization at cell junctions and lamellipodial-like protrusions of the EVL basal surface at EVL cell borders (arrowheads in right panel). Scale bar, 250 μm (**a**) and 25 μm (**b,d,e**).

role of mechanical cues in the process. How local relaxation in tension was transmitted across the epithelium is yet to be determined, but as cell number can not be modified in the EVL due to failed cytokinesis, tissue relaxation possibly involves changes in cell shape, volume and/or reorganization as seen in other epithelia under geometrical constraints[8,9,20]. Together, these results show that modulation of EVL tension directs DCL spreading and supports the idea that increased tension and/or stiffness at EVL cell borders is the main determinant of the stereotyped spatial distribution of DCL cells during epiboly.

**Model of three forces driving DCL spreading.** We have shown that spreading of the embryonic DCL involves a combination of autonomous random motility, a dragging effect by the expanding extra-embryonic surface epithelium (the EVL) and short-range attractive interactions with EVL cell borders. To test if these three

components are sufficient to drive DCL migration and clustering along EVL cell borders in a global embryo context we generated a physical model based on interacting point particles confined to move on the surface of a sphere (Fig. 9a,b and Methods). Numerical simulations considering either a single component or a combination of two forces failed to mimic the experimental conditions (Supplementary Fig. 11). In contrast, simulations that considered the three forces recapitulated the dynamics and spatial configuration of both EVL surface expansion and DCL spreading observed in the experiments, supporting the sufficiency of the model (Fig. 9c–g; Supplementary Fig. 11; Supplementary Movie 9). The physical model also allowed to establish the individual contribution of autonomous and non-autonomous forces to DCL spreading by reducing/removing single components without adjusting the other parameters. By doing this, we found that the principal determinant of vegetal-ward directed DCL spreading was the non-autonomous dragging by the EVL.

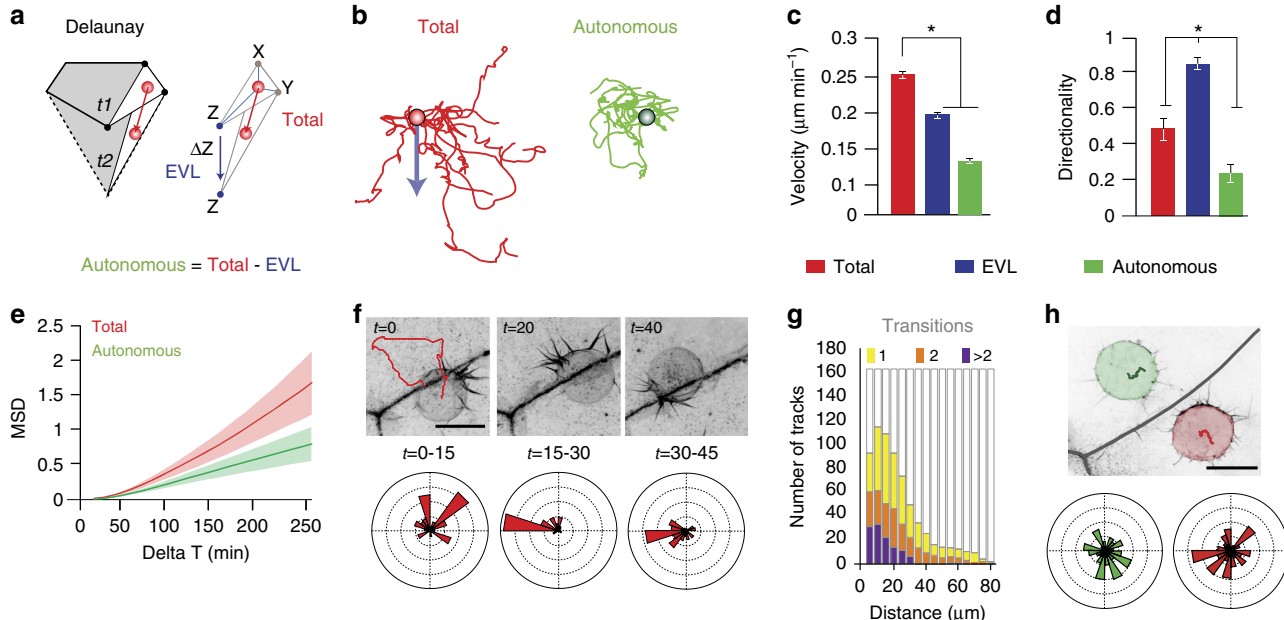

**Figure 6 | Autonomous motility during spreading of the embryonic DCL. (a)** Schematic diagram of the method used to estimate the autonomous movement of DCL cells (see Methods), by subtracting the movement of the EVL from the total DCL cell movement. **(b)** Examples of tracks for autonomous and total DCL cell movements, starting at a common point and aligned to the movement of EVL cells (blue arrow; $n = 8$ tracks). **(c,d)** Quantification of the mean velocity **(c)** and directionality **(d**, ratio of the distance between starting and ending points by the actual trajectory) of cell movement in the three depicted conditions. Values are expressed as means ± s.e.m. *$P < 0.05$ (Wilkoxon). **(e)** Quantification of the mean square displacement (MSD) for autonomous and total DCL cell movements, expressed as means ± s.e.m., as a measure of the area explored by cells for any given time interval. Plots from **b–d** correspond to 9 cells followed over a period of 18 h (54 to 72 h.p.f.), extracted from Supplementary Movie 1. **(f–h)** DCL migratory behaviour at EVL cell borders. **(f)** Confocal microscopy z-stack maximum projections of an embryo expressing *lifeact-GFP*, showing a single DCL cell moving around the EVL cell border during a period of 45 min. The red line corresponds to the track followed by the cell barycentre. Circular plots depict the orientation of filopodial-like membrane protrusions during three consecutive 15-min periods (from Supplementary Movie 3). **(g)** Distribution of the number of tracks of DCL movement that crossed defined boundary regions at growing distances from EVL cell borders, as a measure to define the preferential region of DCL migration. Different colours depict different number of transitions (from Supplementary Movie 1). **(h)** In embryos expressing *lifeact-GFP* and overexpressing Rac1-T17N, DCL cells barely move at EVL cell borders (Supplementary Movie 4). Red and green lines depict the tracks followed by cells during a period of 20 min. Circular plots depict the orientation of thin filopodial-like membrane protrusions of the two DCL cells. Scale bar, 30 μm **(f)**.

The reduction/removal of this force from simulations severely affected cell dispersion (Fig. 10a–d) and reduced the total area covered by the DCL (Fig. 10e,f). Autonomous random motility, on the other hand, facilitated DCL spreading by increasing both the area explored by cells (Fig. 10b; Supplementary Fig. 12) and the distance among cells (Fig. 10a and i—top), and by preventing the stabilization of DCL cells at EVL cell borders (Fig. 10c and i— bottom). Finally, EVL cell border attraction appeared dispensable for DCL cells to disperse and cover the egg surface (Fig. 10a,d and k—top; see also Supplementary Fig. 11). However, it was fundamental to spatially organize the distribution of DCL cells along EVL cell borders (Fig. 10c and k—bottom). Strikingly, the configurations predicted by the independent reduction/removal of EVL dragging and autonomous random motility were experimentally mimicked by abrogation of E-cad function (dn E-Cad or *e-cad*-MO; Fig. 3e–g; Supplementary Figs 4 and 5) and disruption of Rac1-mediated polarized cell protrusions (Rac1-T17N) (Figs 6h and 10j,m; Supplementary Movie 4), respectively. On the other hand, experiments of mild knock down of E-cad function through the injection of small doses of *e-cad*-MO replicated the simulation result predicted by removal of short-range attractive interactions with EVL cell borders. In this condition, the preferential distribution of DCL cells along EVL cell borders was lost without affecting the total area covered by the DCL (Fig. 10l,m; Supplementary Movie 10). This finding highlights that short-range EVL cell border attraction can be experimentally dissociated from EVL dragging and thus works as an independent force that controls a particular aspect of DCL spreading.

Such force requires E-cad function and serves to constrain autonomous random DCL cell movements along structural patterns of the cellular substrate.

## Discussion

Biochemical signals from extra-embryonic tissues direct embryo patterning and coordinate morphogenesis during early vertebrate development. For example, the hypoblast in chick, the anterior visceral endoderm in mouse and the YSL in zebrafish produce signals to position the site of gastrulation (reviewed in ref. 21). Also, the hypoblast coordinates cell signalling and movements of the epiblast in the pre-gastrula chick embryos[22], while ECM proteins derived from the mouse trophectoderm/visceral endoderm and zebrafish YSL are required for morphogenesis of the egg cylinder[23] and heart[24], respectively. In this study, we provide an additional layer of extra-embryonic control of embryo morphogenesis that couples the adhesive and tensile properties of adjacent tissues in the early vertebrate embryo.

The extra-embryonic EVL of teleosts is a surface squamous epithelium thought mainly to provide protection to the embryo and give rise to a small population of cells involved in left–right patterning[25,26]. Here we reveal that the EVL also directs morphogenesis of embryonic tissue during early development. This function relies on the ability of embryonic cells to couple their autonomous random motility to non-autonomous mechanical signals arising from the epibolic expansion of the EVL, which is used by embryonic cells as a substrate for

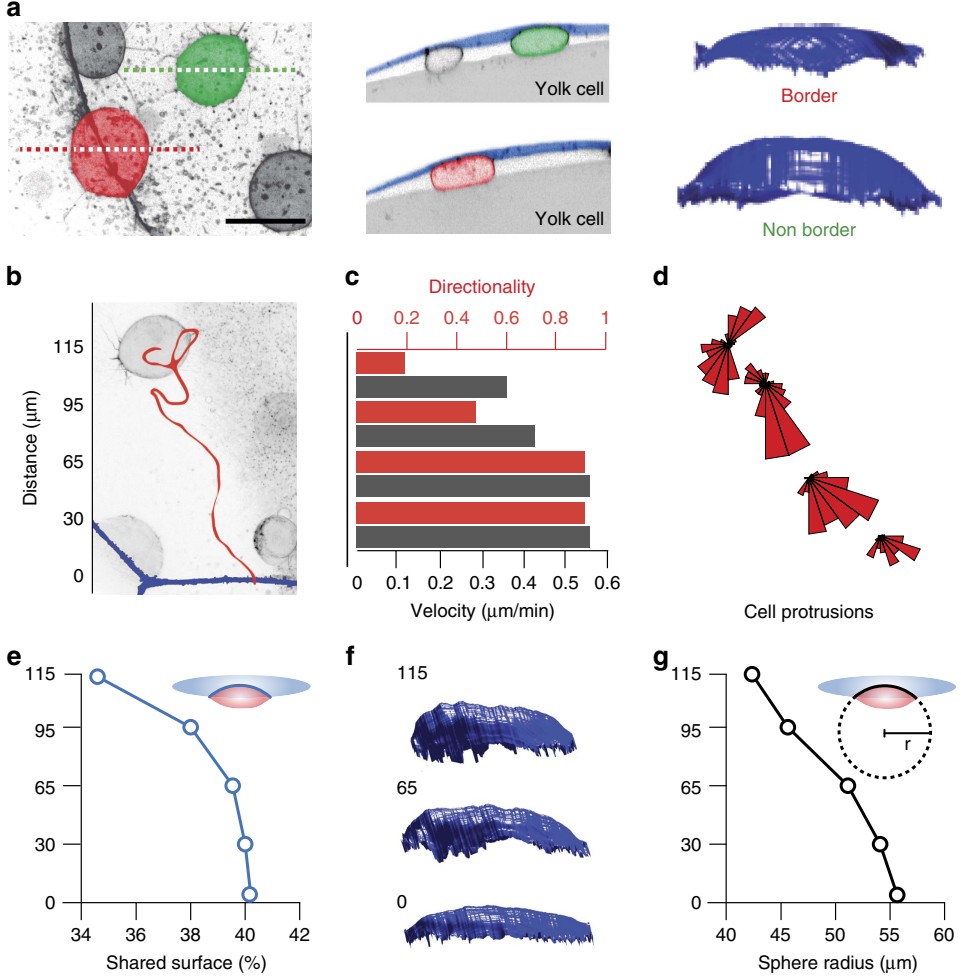

**Figure 7 | Contact interactions and migratory behaviour at and away EVL cell borders.** (**a**) Confocal microscopy z-stack maximum projection images with an inverted look-up table (left) and orthogonal optical sections along the depicted dashed lines (middle) of a 60% epiboly *A. nigripinnis* embryo expressing *lifeact-GFP*. DCL cells pseudocoloured in red and green are representative examples obtained from the analysis of *n* > 3 embryos of cells located at and away from the EVL cell border, respectively. 3D volumetric projections of the DCL-EVL contact surface corresponding to the coloured DCL cells are shown in the right panels. (**b–g**) Changes in migratory behaviour, shared surface and EVL basal surface deformation as the DCL cell approaches the EVL cell border. (**b**) Confocal microscopy z-stack maximum projection of an embryo expressing *lifeact-GFP*, showing a single DCL cell that is about to move towards the EVL cell border (blue line at the bottom). The red line corresponds to the track followed by the cell as it approaches the border during a period of 45 min. Numbers on the left indicate the distance to the EVL cell border (distance = 0), used to define intervals for the quantifications shown in (**c–g**). (**c**) Quantification of the velocity (dark brown) and directionality (red) of DCL cell movement. (**d**) Circular plots depicting the orientation of filopodial-like membrane protrusions of the DCL cell shown in **b**. (**e**) Quantification of the DCL-EVL contact surface (shared surface). (**f**) 3D volumetric projections of the DCL-EVL contact surface at defined distances from the EVL border (numbers on left). (**g**) Quantification of the curvature radius (*r*) of the sphere that fits into the 3D volumetric surface area shared between the DCL cell and the EVL basal surface, as a measure of EVL basal surface deformation. (**b–g**) were obtained after analysis of Supplementary Movie 6.

migration. Tissue-tissue mechanical coupling requires E-cad-dependent cell–cell adhesive and tensile interactions that result in both dragging of embryonic cells by the vegetal movement of the EVL and short-range attraction towards regions of increased cortical tension and/or stiffness, the EVL cell borders. Together, autonomous motility, dragging and short-range attraction forces direct cell migration and the spreading of embryonic tissue following structural features of the extra-embryonic cellular substrate in a way reminiscent to contact guidance[27,28].

We foresee that the coupling of mesenchymal-like tissues to adhesive and tensile properties of adjacent epithelia might represent a fundamental cellular principle of mesenchymal-like tissue spreading in developmental contexts and pathological conditions of increased tension, for example during the closure of epithelial wounds and the migration of cancer cells. We also predict that upon this primary principle, more elaborate patterns

of tissue spreading can emerge as a consequence of increased cell density, synchronous patterning signals, or concomitant morphogenetic events. Indeed, in epiboly of *Fundulopanchax gardneri*, a semi-annual killifish species that contains a larger number of DCL cells compared to annual killifish, we observe a transient phase of radial cell intercalation that precedes the formation of a cell monolayer, where DCL cells show a preferential distribution towards EVL cell borders (Supplementary Fig. 13). The situation in zebrafish is more complex, as cells of the DCL form several layers and exhibit extensive radial cell intercalation while undergoing concurrent movements of epiboly and axis formation[7]. Notably, DCL cells display random motility[29] and can use the EVL basal surface as a substrate for adhesion[30–33]. These findings suggest a common role of the extra-embryonic EVL in directing the spreading of the DCL in different teleost species, and opens the intriguing

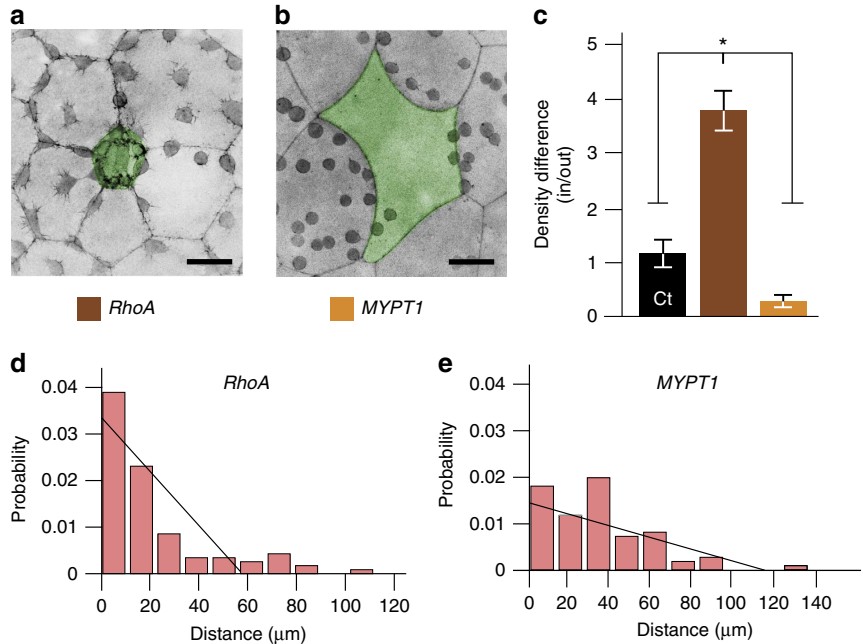

**Figure 8 | Role of mechanical input from the extra-embryonic EVL in directing DCL spreading.** (**a,b**) Confocal microscopy z-stack maximum projections with an inverted look-up table, showing a central EVL cell (pseudocoloured in green) overexpressing *rhoA* and *N-ter-MYPT1* (*MYPT1*). RhoA-mediated increase in cortical tension limits the surface expansion of the EVL cell and renders it smaller than neighbouring EVL cells (**a**). MYPT1-mediated decrease in cortical tension increases the surface expansion of the EVL cell rendering it larger and more concave, compared to neighbouring EVL cells (**b**). (**c–e**) Effect of the local modulation of EVL tension on the spatial distribution of DCL cells. (**c**) Quantification of the ratio of DCL cell density (number of DCL cells under the EVL per given area) between the injected EVL cell ('In'; green cells in **a** and **b**) and surrounding EVL cells ('Out'). Values are expressed as means ± s.e.m., and correspond to control and injected conditions (*RhoA* and *MYPT1*, as indicated) (*n* = 3 embryos per condition). *P < 0.01 (Wilcoxon). (**d,e**) Probability distribution of DCL cell position as a function of the distance to EVL cell borders (measured from the DCL cell centre of mass to the closet EVL cell border) for the *RhoA* (**d**) and *MYPT1* (**e**) conditions shown in **a** and **b**. The black straight lines corresponds to the expected random distributions for the EVL cell surface radius observed in both experimental conditions. Scale bar, 100 µm (**a,b**).

possibility that radial cell intercalation might emerge as a consequence of the mechanical coupling that the DCL establishes with an expanding EVL during epiboly. Future works including more direct mechanical perturbations and measurements will have to test this hypothesis directly.

## Methods

**Fish maintenance and husbandry.** Adult wild type *Austrolebias nigripinnis* (*A. nigripinnis*) were raised and maintained following the bioethical guidelines determined by the Ethics Commission of the Faculty of Medicine, Universidad de Chile. Fish were kept under 12/12 h(h) light/dark cycle regime at 17–20 °C, in 40 litre aquaria with sponge filtering (Aquarium-Schwammfilter) and water aeration. Fish water of 150 µS conductivity was prepared by adding 140 g microsalt (Brustmann), 140 g Mineral Salt (Sera) and 4 drops of liquid ferrogan fluid (Hobby) to 40 l of reverse osmosis water, adjusting pH to 7–7.5 by adding sodium bicarbonate. In each tank, fish were kept at a male to female ratio of 1:1 to 1:3, and fed 3 times per day with newly hatched brine shrimps and freshwater live food (*Eisenia foetida*, *Lumbricus variegatus*). Weekly, 50% of total water volume was replaced with fresh water to remove the waste produced by fish and the supplemented food. For breeding, plastic boxes filled with peat moss were placed at the bottom of tanks, wherein eggs could be laid. Fertilized eggs were recovered manually by placing the peat moss on a filter paper, and transferred to Petri dishes containing ERM rearing medium (NaCl 17.1 mM, KCl 402 mM, CaCl₂ 272 mM, MgSO₄ 661 mM, pH 6.3). Embryos were kept at 25 °C and staged according to hours (h.p.f.) and days (d.p.f.) post-fertilization.

**Embryo microinjection.** Microinjection of mRNAs and morpholino antisense oligonucleotides (MOs) were performed in embryos of *A. nigripinnis* at one-cell, two-cell, four-cell and late blastula (48 h.p.f.) stages. One-cell stage microinjection and microinjection of both blastomeres at two-cell stage resulted in homogeneous distribution of mRNA and MOs, respectively. Microinjection of two blastomeres at four-cell stage resulted in mosaic mRNA expression. Microinjection at late blastula allowed the assessment of mRNA expression in a single EVL cell during epiboly. For microinjection, embryos were placed in a petri dish previously covered with a layer of agarose and containing ERM rearing medium. Volumes between 500 pL and 1 nl were microinjected by applying pressure using a picospritzer (IM 300 Cell

microinjector, Narishige). Microinjection was performed by inserting the tip of the micro-needle directly into the cell, under the control of a manually driven micro manipulator (Brinkmann Instruments). Microneedles were made of glass capillaries (1B100F-6 World Precision Instruments) and pulled in a horizontal puller (Model PC-86 from Sutter instruments) to reach a pipette shape similar to those used for microinjection of medaka (*Oryzias latipes*) embryos[34]. After injection, embryos were maintained in the same petri dish at 25 °C in ERM rearing medium. For mRNA synthesis, pCS2⁺ expression vectors containing cDNAs for *GAP43-EGFP*[35], *h2b-RFP*[36], *lifeact-GFP* and *lifeact-mCherry*[37], *rac1-T17N*[38], *N-ter-MYPT1* (ref. 39), *rhoA*[40] and dominant-negative *e-cadherin* (see below) were linearized and *in vitro* transcribed using mMessage-mMachine kit (Ambion) following standard protocols.

**Total RNA extraction and RT–PCR.** Trizol Reagent (Invitrogen) was used to extract total RNA from embryos at different stages of development. Tubes with up to 10 embryos were kept in liquid nitrogen until processing. Embryos were grinded with a plastic micropestle as much as possible to ensure a complete tissue desegregation. Then, the pestle was lifted slightly, 400 µl Trizol Reagent added, and the homogenate allowed to thaw. Before removal, the pestle was washed with additional 100 µl Trizol to recover any material stuck to the pestle. The homogenate was briefly mixed with a vortex and kept at room temperature (RT) for 5 min to allow nucleoprotein complex dissociation. After, the homogenate as a whole was transferred to pre-prepared phase-lock gel heavy containing tube (MaXtract High Density, Qiagen). Subsequently, 100 µl chloroform was added and the mixture, shook by hand vigorously for 15 s(sec) and kept at RT for 3 min. The tube was centrifuged for 15 min at 12,000 × g and the aqueous phase transferred to a new 1.5 ml eppendorf tube. Taking into account the small size of the sample, 20 µg of RNase-free glycogen (Invitrogen) was added as a carrier to the aqueous phase. RNA precipitation was started by adding 250 µl of isopropanol and incubated at RT for 10 min. The sample was centrifuged for 10 min at 12,000 × g, the pellet washed once using 1 ml of 75% ethanol, and air-dried. RNA was resuspended in 10 µl nuclease-free water (GIBCO). The amount of RNA per µl was measured by determining the spectral absorbance at 260 nm, and the integrity of the RNA verified in agarose gel. Total RNA (0.5–1 µg) was reversely transcribed to produce cDNA using Superscript III reverse transcriptase (Invitrogen) primed with Oligo (dT)12–18 (Invitrogen).

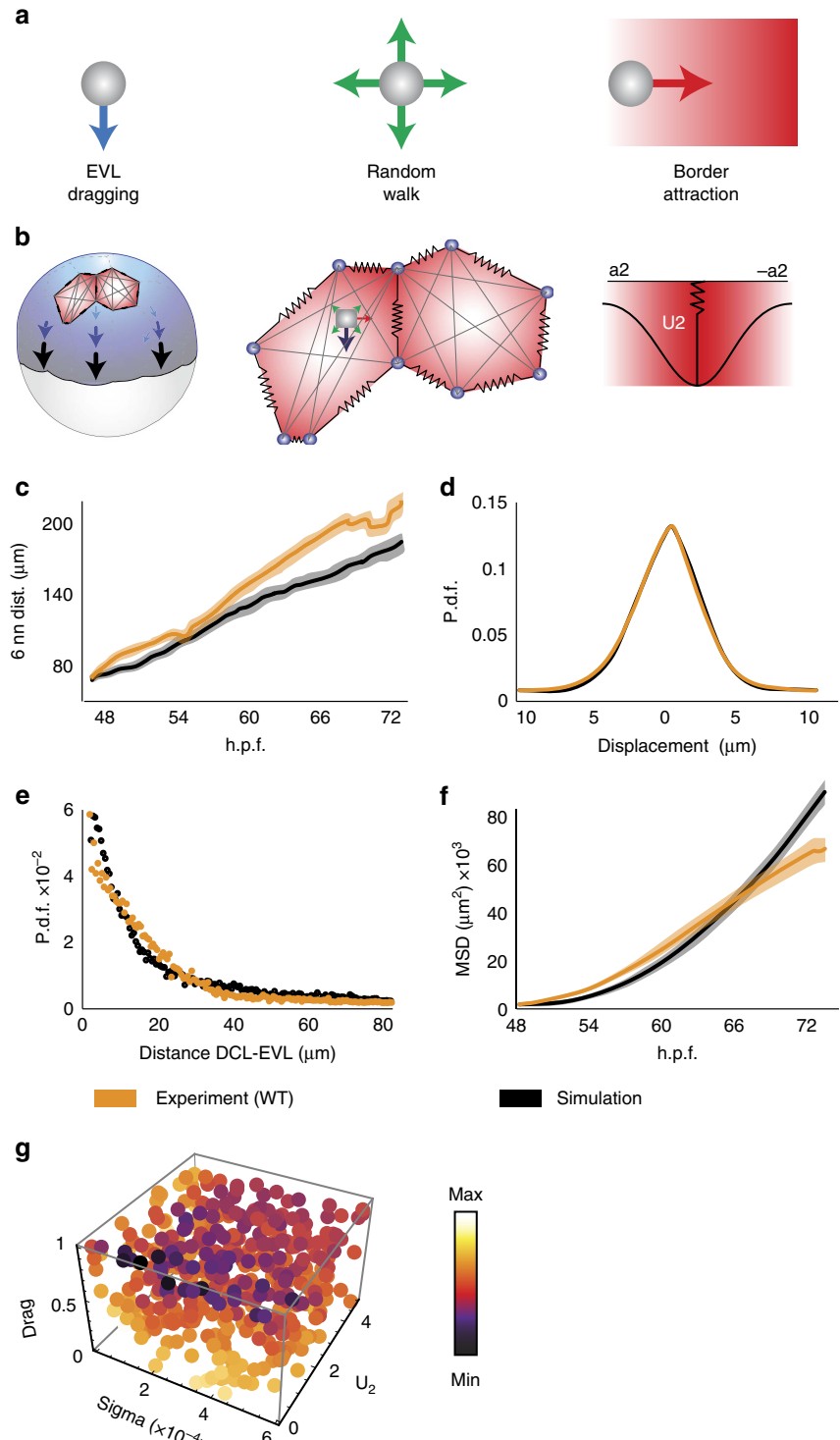

**Figure 9 | Model of three forces driving DCL spreading.** (**a**) Proposed physical model of three forces driving spreading of the embryonic DCL. (**b**) Schematic diagram of the simulation model, based on interacting point particles (EVL vertices, DCL cells) confined to move on the surface of a sphere. EVL vertices (blue circles) are connected by springs that move and expand under a constant velocity from the EVL epithelial margin (border condition). DCL cells (grey circles) move with an autonomous random walk, and are influenced by non-autonomous forces of EVL dragging and short-range EVL cell border attraction. Attraction is determined by a gaussian potential along EVL cell borders with length $a_2$ and amplitude $U_2$ (see Methods). (**c–f**) Comparison of four functions of DCL spreading between simulations (black lines) and experimental data (orange lines) (Supplementary Movie 9). The mean distance to the six nearest neighbours within the DCL (**c**) is a measure of cell dispersion. The probability density function of the displacement of DCL cells (**d**) is a measure of noise. The probability distribution of the distance of DCL cells to the nearest EVL cell border (**e**) is a measure of the spatial segregation of DCL cells to EVL cell borders. The mean square displacement (**f**) is a measure of the area explored by cells for any given time interval. (**g**) Sampling of the free parameters $D$ (drag), $\sigma$ (sigma) and $U_2$. Colours indicate the normalized values of the target functions (according to the colour scale on the right) for different values of the free parameters. Minimum values concentrate around drag $\sim 1$, $\sigma \sim 2 \times 10^4$ and $U_2 \sim 0.3$.

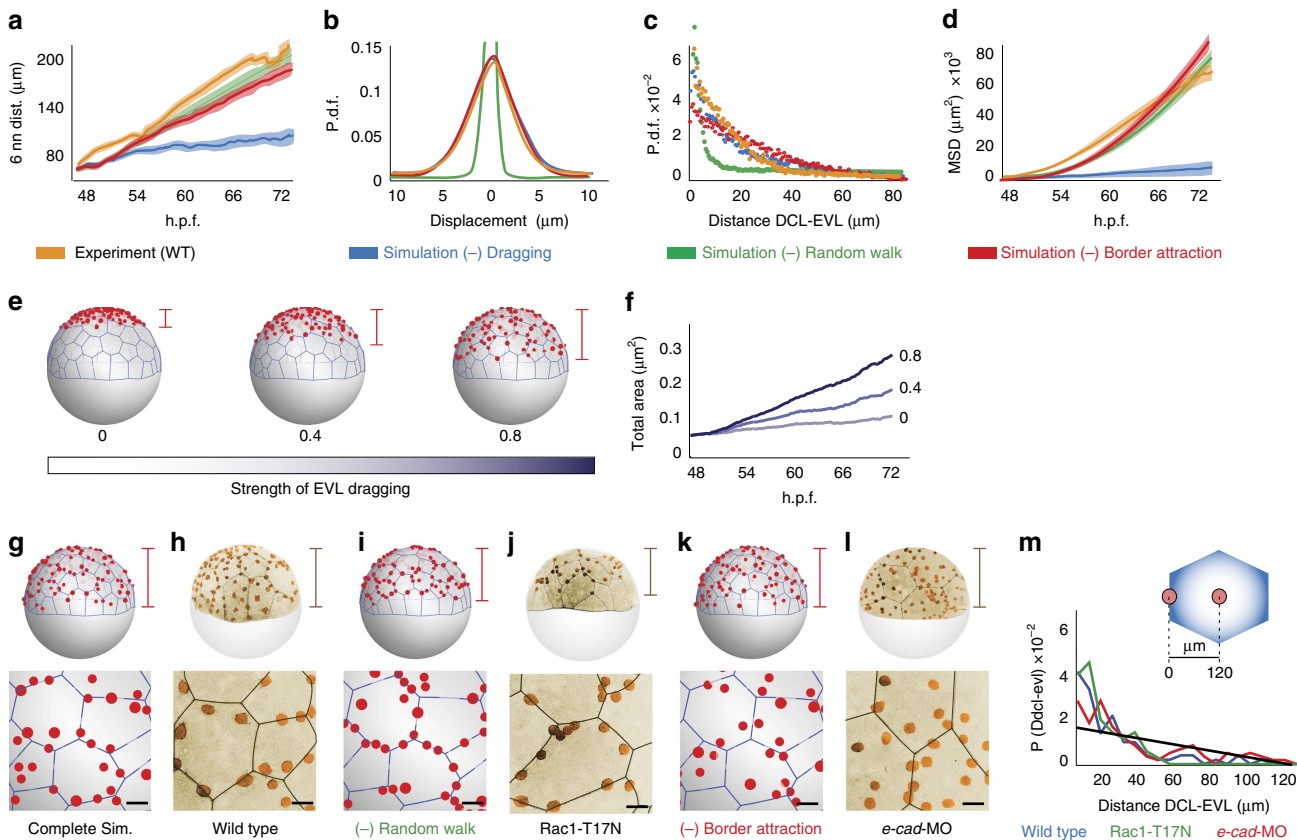

**Figure 10 | Testing the role of autonomous and non-autonomous forces of DCL spreading.** (**a–d**) Effects of the independent removal from simulations of autonomous random walk (green), EVL dragging (blue) and EVL cell border attraction (red), on the dynamic functions of DCL spreading defined in Fig. 9c–f (**e,f**). Effects of the gradual reduction of EVL dragging from simulations in the vegetal advancement of the DCL margin (**e**) and the total area covered by the DCL (**f**). The grayscale bar in (**f**) indicates the strength of EVL dragging (ranging from 0 – no drag, to 1 – maximum drag value as in simulations from Fig. 9c–f). (**g–m**) Comparison of results obtained from simulation predictions and experimental conditions that differentially affect random walk and EVL cell border attraction. In **g–l**, the vegetal advancement of the DCL is shown in the top panels (brackets) while the distribution of DCL cells with respect to EVL cell borders is shown at the bottom. (**g,i,k**) Effects of the independent removal from simulations of random walk (**i**) and EVL cell border attraction (**k**). (**h,j,l**) Phenotypes induced by experimental disruption of Rac1-mediated polarized cell protrusions (overexpression of Rac1-T17N) (**j**) and mild morpholino-mediated knock down of E-cad function (**l**). (**m**) Probability distribution of DCL cell position as a function of the distance to EVL cell borders in the conditions depicted in **h,j** and **l**. The black straight line corresponds to the expected random distribution for a mean EVL cell surface radius of 120 μm, as indicated in the top right corner. Scale bar, 50 μm (bottom **g–l**).

**Genomic DNA extraction.** For genomic DNA extraction, one or two embryos were transferred to a microfuge tube, the embryo medium removed with a pasteur pipette and 100 μl of lysis buffer (10 mM Tris pH 8.2, 10 mM EDTA pH 8, 200 mM NaCl, 0.5% SDS, 200 μg ml$^{-1}$ Proteinase K) was added. Embryos were then incubated at 50 °C for at least 3 h with occasional gentle swirling. The mixture was cooled at RT, extracted with one volume of phenol:chloroform:isoamyl alcohol (25:24:1) and centrifuged for 10 min at 5,000 × g. The upper phase was carefully removed and DNA precipitated by adding 1:10 volume of sodium acetate pH 6 and two volumes of ethanol. The precipitated DNA was removed by using the pasteur pipette and transferred into a tube containing 70% ethanol. The DNA pellet was stood in 70% ethanol for about 5 min and gently moved it around from time to time using the pasteur pipette. Finally, the DNA was removed from the 70% ethanol with a pasteur pipette, letting the excess liquid to drip off, and placing the pasteur pipette with DNA sticking to it inverted into a microfuge rack. DNA was air dried for 5 min and resuspended in TE buffer.

**Cloning of *A. nigripinnis* E-cadherin.** Degenerate primers for *e-cadherin/cadherin-1* (*e-cad*) were designed using conserved nucleotide regions within genes as targets. Sequence alignments were performed using ClustalW software, and *e-cad* gene orthologs belonging to different teleost fishes, including *Tetraodon nigroviridis* (ENSTNIG00000003590), *Danio rerio* (ENSDARG00000102750), *Takifugu rubripes* (ENSTRUG00000002439) and *Oryzias latipes* (ENSORLG00000020295), were used as inputs. The primer sequences used were: forward-*e-cad*: 5′-GACAACTCTGATATCCGCTACC-3′, reverse-*e-cad*: 5′-TGTTATCTCTGGTGTCATCGC-3′. PCR products were gel purified using AxyPrep DNA Gel Extraction Kit (Axygen) and cloned into pCRII-TOPO cloning vector (Invitrogen) according to manufacturer's instructions. The identity of cDNA fragments was confirmed by sequencing and conserved protein prediction using the conserved

domain database from NCBI (http://www.ncbi.nlm.nih.gov/Structure/cdd/wrpsb.cgi).

**E-cadherin knock-down by morpholino antisense oligonucleotides.** Splice-inhibiting MOs were obtained from Gene Tools, maintained in nuclease-free water at a final concentration of 1 mM, and stored at − 80 °C. MO sequences used were: *e-cad*-MO, 5′-CATCATTTTCAAGTTCTTACGTCAA-3′; Control-MO, 5′-CCTCTTACCTCAGTTACAATTTATA-3′. A suitable target exon-intron sequence located in the cytoplasmic region of the gene was obtained after analysis of genomic exon-intron organization of *A. nigripinnis e-cad* by PCR amplification, cloning and sequencing of several genome regions. Variable volume of MOs at stock concentration were injected directly into both blastomeres at two-cell-stage, to reach 6–8 ng MO per embryo. Analysis of knock down efficiency was determined by PCR amplification. Anomalous splicing induced by MO injection was confirmed by cloning and sequencing of the corresponding PCR-obtained gel bands.

**Expression of dominant-negative E-cadherin.** The dominant-negative (DN) *A. nigripinnis* E-cad was designed as previously reported for zebrafish[14]. Briefly, primers were designed to obtain a PCR fragment that included the complete Cadherin sequence from the extracellular domain (ECD) 3 to the cytoplasmic domain, therefore excluding the ECD 1 and 2 (known to be critical during the first recognition step). The fragments were cloned into pCS$^{2+}$ vectors using EcoRI and XhoI restriction sites and the identity confirmed by sequencing. Plasmids were linearized with NotI, and mRNA obtained using mMessage-mMachine kit (Ambion) according to manufacturer's instructions. For homogenous embryo expression, 1-cell stage embryos were microinjected with 100–180 pg mRNA per embryo. For mosaic expression, 2 blastomeres of four-cell stage embryos were

microinjected with mRNA to reach total 100–180 pg per embryo. As dn E-cad was not tagged, *GAP43-EGFP* mRNA was added to the mRNA mix and the intensity of GFP fluorescence was then used as a measure of dn E-cad expression.

**Whole-mount *in situ* hybridization.** Whole-mount *in situ* hybridization was performed using reagents obtained from Roche. Following overnight fixation in 4% paraformaldehyde-PBS at 4 °C, chorions were removed from embryos by hand using forceps. Antisense RNA probes for *A. nigripinnis e-cad* were synthesized using the specific partial coding sequences obtained from cloning and labeledwith UTP-digoxigenin. Embryos were incubated at 70 °C in hybridization solution containing 50% formamide. Probes were detected using alkaline phosphatase conjugated antibodies and visualized by 5-bromo-4-chloro-3-indolyl-phosphate (BCIP) staining. Sense labelled probes were used as controls. Images were obtained using Volocity ViewVox spinning disc (Perkin Elmer) coupled to a Zeiss Axiovert 200 confocal microscope.

**Indirect immunofluorescence.** Embryos were freshly collected and fixed overnight at 4 °C in 4% PFA in PBS, washed 3 times in wash buffer (PBS, Triton X-100 0.5%), manually dechorionated and blocked for 2 h in blocking buffer (PBS, Goat serum 10% and Triton X-100 0.5%) at RT. The blocking step, and all following steps, were performed under soft constant shaking. Primary antibodies used were anti-phospho Myosin Light Chain (1:100, Cell Signaling #3671) and custom-made rabbit-derived antiserum against E-cadherin (1:50, Strategic Diagnostics Inc., SDIX). The chosen epitope for recognizing the E-cadherin protein was: NH2-CFRNDVAPAFMPAPQYRPRPA-amide (cytoplasmic domain). Antibodies and antiserum were diluted in blocking buffer and embryos incubated overnight in this solution at 4 °C. Subsequently, embryos were washed three times in washing buffer and incubated in 1:200 goat anti-rabbit Alexa Fluor 488 (Molecular Probes) solution in blocking buffer overnight at 4 °C. For concomitant F-actin staining, Phalloidin-Alexa 568 (Molecular probes) at 1:200 was added to the solution. The embryos were then washed 3 times and kept in PBS until mounting for imaging.

**Confocal scanning and spinning-disk microscopy.** Embryos of *A. nigripinnis* were mounted in 1% low-melting-point agarose in either PBS (fixed immunostained embryos) or ERM rearing medium (living embryos) in a custom designed chamber, and placed on the microscope stage. For *in vivo* imaging, the temperature was kept constant at 25 °C throughout the experiment using a temperature control system. Whole embryo dynamic *in vivo* imaging was performed in a Leica TCS LSI Confocal microscope with HCS software using a ×5 objective with 10× optical zoom and 488/520 (excitation/emission wavelengths) lasers. For high temporal and spatial resolution at cellular and sub-cellular levels, embryos were imaged in a Volocity ViewVox spinning disc (Perkin Elmer) coupled to a Zeiss Axiovert 200 confocal microscope using a Plan-Apochromat ×40/1.2 W or a Plan-Neufluar ×25/0.8 W objective with lasers 488/520, 568/600 and 647/697 nm (excitation/emission wavelengths). Processing and analysis of digital images were performed using Fiji[41], Matlab[42], Volocity (Improvision) and Adobe photoshop.

**Shared surface estimation and visualization of cell contact area.**
Representative high-resolution spinning disk microscope z stacks of *A. nigripinnis* embryos expressing *GAP43-EGFP* or *lifeact-GFP*, taken parallel to the embryo surface at 0.5 μm interval, were chosen to quantify the contact area between the membrane of individual cells of the deep cell layer (DCL) and enveloping cell layer (EVL). For this, orthogonal views along xz and yz axes at 0.166 μm intervals were generated and used to manually draw lines depicting the DCL-EVL contact zones using a custom-made routine in Fiji[41]. Then, Matlab routines[42] were used to estimate the surface mesh, using a smoothed mesh by Gaussian and bilateral filters, followed by 3D mesh reconstruction and surface quantification using 3D surface reconstruction techniques with colour interpolation and Phong reflection model. We estimated the radius fitting the best sphere minimizing the equation $x^2 + y^2 + z^2 + ax + by + cz + d = 0$ in the mesh 3D points.

**Cell segmentation.** Images obtained from whole embryo *in vivo* imaging comprised a $1,024 \times 1,024 \times 164 \times 146$ (*xyzt*) voxel size volume. To simplify cell segmentation, a max z-projection using Fiji[41] was first applied to each *xyz* stack to obtain a 2D sequence of size $1,024 \times 1,024 \times 146$ (*xyt*). The volume was then segmented using a random forest approach[43], also available in FIJI, with active Sobel and Laplacian features. As *GAP43-EGFP* simultaneously labelled the cell membrane of the EVL and DCL, two random forest trainings highlighting cells of these two layers were performed. For EVL cells, a skeleton algorithm[44] reduced segmented membrane thickness to one-pixel-wide to smooth EVL contours. Manual correction by a biologist verified that DCL cells were not in contact to each other and EVL segmented membranes were closed curves. Supplementary Fig. 14a–d shows the 2D segmentation procedure.

**Cell tracking.** An initial custom distance-based tracking over the cell centre of mass was implemented in Matlab for cells of the DCL and EVL[42]. The tracking

algorithm linked a track to the closest cell in the next frame by sequentially searching close candidates. To handle errors in the 2D tracking due to ambiguities (that is, DCL cells reaching the EVL appeared as cell divisions), temporal variations on the cell area (200% for EVL and 50% for DCL), speed (35 μm per frame), or appearing/disappearing trajectories were reported to the biologist as potential errors for manual corrections of segmentation or tracking. The two mentioned processes, segmentation and tracking, were in a 2D projection of the 3D stack. To estimate the z-component of the centre of mass or any other point of the cells, the maximum z-intensity at each *xyt* location was searched. This depth approximation drastically reduced the amount of manual segmentation and tracking work but it had the disadvantage to locate close DCL and EVL cells at the same depth and to deliver noisy z-component values. Noisy depth was handled by a spherical projection explained in the following section.

**Drift correction and sphere projection.** As imaging was recorded in embryos within the chorion, in which they could experience rotational movements within the shallow peri-viteline space, rotation became an important issue to correct specially in long recordings. Translation was not relevant as the egg was fixed in agarose. Additionally, the rotational drift must be handled combined with the normal tissue expansion observed during epiboly. We thus built a growing plus rotational calibration dataset to test drift removal algorithms, choosing the point-to-surface minimizing algorithm proposed by Hunyadi[45] that successfully handled an isotropic cell growing plus rotation scenario (Supplementary Fig. 15). The chosen drift removal algorithm was applied to align the animal–vegetal egg axis to the z-axis in all frames.

To address noisy z-component computed from the segmentation and tracking, we projected *xyz* segmented voxel locations onto a sphere surface by using the sphere equation, that is, retrieving $Z_c$,

$$Z_c = \sqrt{R^2 - (x^2 + y^2)} \tag{1}$$

where (x,y) are voxel locations and R is the sphere radius ($r = 592$ μm) fitted in the last frame using a quadratic target function taking all the points from segmented EVL and DCL cells in 3D. Supplementary Fig. 14 shows the volume segmentation procedure and an estimation of the error generated by measuring distances on the deformed surface relative to real z-axis distance.

**Analysis of cell parameters.** The distance between DCL cells was computed as the angle between the centre of mass position times the egg sphere radius (distance on the sphere surface). DCL-EVL cell edge distance was computed as the shortest distance of the DCL cell centre of mass to the EVL cell membrane. DCL and EVL cell area were computed using a polygonal contour approximation of the perimeters and using a line integral based on Green's Theorem implemented in the sphereint Matlab function[42]. The DCL covered area was estimated by identifying DCL cells in the 2D projected convex hull of the DCL centres, ignoring the z-component, and then computing the area with the described line integral approach. For cell shape parameters, the three principal axes were computed from the segmented image using second-order moments[1]. The first axis defines the orientation of maximum variance and is proportional to the variance magnitude ($\lambda_1$) defining an object length; secondary and tertiary axis are orthogonal and define in the same way width and height with associated sizes $\lambda_2$ and $\lambda_3$, respectively. Elongation is defined as $1 - \lambda_2/\lambda_1$ and flatness as $1 - \lambda_3/\lambda_2$, with both indexes in the interval [0,1]. The entropy index is computed as $-\sum \lambda'_i \log(\lambda'_i)$, where $\lambda'_i = \lambda_i/\sum \lambda_i$, thus it accounts for cell elongation heterogeneity (entropy index is maximum when all indexes are equal). The eccentricity index corresponds to the distance between foci and major axis length of an ellipse with the same second-moments as the cell. Cell trajectory directionality was estimated as the ratio of displacement over trajectory length both in 3D. Nearest neighbour distance was computed by building the distance matrix connecting all DCL cells on the sphere surface and averaging the three closest distances. The mean square displacement for DCL cells was calculated by taking a sliding temporal window (of 1 to 25 frames) and computing the track displacement as the 3D Euclidean distance between initial and final window frames. Linear regression (using Matlab Curve Fitting Toolbox with least-squares option) reporting $r^2$ value was applied to study the relation between the EVL total area and DCL covered area during epiboly. To all parameter pairs, when normally distributed, a *t*-test was used to verify whether mean values differences were statistically significant. For other distributions, a Wilcoxon test was used.

**DCL cell autonomous motion estimation.** DCL cell movement under the EVL was decomposed into a non-autonomous dragging component, largely due to EVL movement to which DCL cells adhere, and an autonomous component due to intrinsic DCL cell motility. The decomposition requires an EVL deformation model to estimate the deformation at any EVL surface point from the available EVL contour information, as the EVL moves and grows during epiboly. Two EVL deformation scenarios were studied: isotropic and anisotropic, and three EVL geometric deformation models were proposed and compared. The compared models were the two closest EVL cell vertices, the EVL cell centre of mass, and the barycentric coordinates defined by the EVL cell vertices Delaunay triangulation over the 2D projection. In both deformation scenarios, the Delaunay-based

geometric model showed the lowest average error (Supplementary Fig. 16), in average underestimating EVL deformation. DCL trajectories were decomposed by subtracting the weighted trajectories of the EVL cell vertices defining the Delaunay triangle to which the DCL was associated to. Weights were barycentric coefficients ($\gamma$) of the DCL position expressed as a function of the 3 EVL vertex positions and recomputed at each frame,

$$\gamma_k \triangleq [\gamma_{k1}, \gamma_{k2}, \gamma_{k3}]^T = [\mathbf{r}_{k1}, \mathbf{r}_{k2}, \mathbf{r}_{k3}]^{-1}\mathbf{V}_k, \tag{2}$$

where $\mathbf{r}_{k1}$ is one of three associated EVL vertices positions and $\mathbf{v}_k$ the coordinates of the DCL position.

**DCL cell protrusion detection and analysis.** In high-resolution DCL cell image recordings, protrusions and cell tracking were quantified by segmenting the DCL cell body followed by segmentation of the DCL cell membrane. Cell protrusions were identified by combining body and membrane information. Body segmentation started with the $xyzt$ stack by a median $z$-projection, normalizing brightness (saturation at 0.4) and binarizing the volume by an automatic threshold function (Otsu), all using FIJI[41]. Membrane segmentation was achieved by estimating $z$ position where DCL cell protrusions were observable as function of $xy$ with a quadratic function. The quadratic function was estimated by manually adjusting 9 control points every 10–50 frames, depending on DCL cell movement, in a custom Matlab[42] code. To identify protrusions, objects larger than 1.1 $\mu m^2$ in the cell membrane segmentation in contact with cell body were classified as elongated protrusions with an eccentricity > 0.97, otherwise a solidity parameter > 0.65 (ref. 42) determined round protrusions and else objects with multiple elongated protrusions. After elongated protrusions detection, the base position defined orientation with respect to the cell centre of mass.

**Physical modelling.** Models of interacting point particles confining to move on a sphere were used to describe the behaviour of DCL and EVL cells. The vertices of EVL cells were represented by point particles interacting with other particles/vertices belonging to the same EVL cell. DCL cells were described as noisy point particles that interact among them with a repulsive potential in order to avoid overlapping. As the EVL is the substrate for DCL cell movement, the model included a 'drag' term that took into account such tissue-tissue interaction. Finally, preferential interaction of DCL cells to EVL cell borders was described as a gaussian potential centred along the EVL cell border (Supplementary Fig. 17). We compared experimental movies based on *in vivo* imaging to numerical simulations of interacting point particle models and obtained the values of free parameters of the models that yielded the best agreement with the experimental movies (see below).

**Model of EVL cell expansion.** To model the movement of EVL cells, we considered two populations of EVL cell vertices: (i) vertices that form part to the bulk of the EVL epithelium, and (ii) vertices that belong to the free (vegetal) border of the EVL. We modelled the movement of bulk EVL cell vertices through point particles interacting with neighbouring vertices belonging to the same EVL cell. We considered two types of interactions: among neighbouring vertices that form the edges of a defined EVL cell, and among opposites neighbouring vertices within a defined EVL cell. The latter was considered to avoid an over expansion of each EVL cell. We modelled both types of interactions with an harmonic potential, denoted by $U^{evl}$, with the same elastic constant $U_0$. The equations that modelled the behaviour of the bulk EVL cell vertices were:

$$\frac{d\mathbf{r}_i}{dt} = \mathbf{F}_i^{evl} \tag{3}$$

$$\mathbf{F}_i^{evl} = \sum_{j nn i}\boldsymbol{\nabla}_i U^{evl} + \sum_{j op i}\boldsymbol{\nabla}_i U^{evl} \tag{4}$$

$$U^{evl} = \frac{1}{2}U_0\left(|\mathbf{r}_i - \mathbf{r}_j| - l_{ij}\right)^2 \tag{5}$$

$$l_{ij} = |\mathbf{r}_i(t=0) - \mathbf{r}_j(t=0)| \tag{6}$$

In the case of the vertices from the free border of the EVL, we imposed a velocity with constant magnitude $V_0$ and with tangential direction in the position of the vertex pointing to the south pole $\mathbf{t}$. The equation for these vertices was

$$\frac{d\mathbf{r}_i}{dt} = V_0\mathbf{t} \tag{7}$$

$$\mathbf{t} = (\cos\theta\cos\phi, \cos\theta\sin\phi, -\sin\theta) \tag{8}$$

where $\theta$ and $\phi$ are the angular spherical coordinates.

**Model of DCL cell movement.** DCL cell movement was described by an interacting noisy point particle model with the equation of motion equation (9). The force to avoid overlapping among DCL cells is the first term in equation (9), computed with the potential of equation (12). The second term models the attraction of DCL cells to the borders of the EVL cells, computed with the potential

of equation (13).

$$\frac{d\mathbf{r}_i}{dt} = \mathbf{F}_i^{dcl} + \mathbf{F}_i^{dcl-evl} + D_r\langle\mathbf{v}^{evl}\rangle_i + \boldsymbol{\eta}_i \tag{9}$$

$$\mathbf{F}_i^{dcl} = \sum_{j nn i}\boldsymbol{\nabla}_i U^{dcl}(|\mathbf{r}_i - \mathbf{r}_j|) \tag{10}$$

$$\mathbf{F}_i^{dcl-evl} = \sum_{j nn i}\boldsymbol{\nabla}_i U^{dcl-evl}(|\mathbf{r}_i - \mathbf{r}_j|) \tag{11}$$

$$U^{dcl}(|\mathbf{r}_i - \mathbf{r}_j|) = U_1\left[\left(\frac{a_1}{|\mathbf{r}_i - \mathbf{r}_j|}\right)^2 - 1\right] \tag{12}$$

$$U^{dcl-evl}(|\mathbf{r}_i - \mathbf{r}_j|) = -U_2 a_2^2 \exp\left(\frac{|\mathbf{r}_i - \mathbf{r}_j|^2}{2a_2^2}\right) \tag{13}$$

$$\left\langle \eta_i^p(t)\eta_j^q(t')\right\rangle = \sigma\delta_{ij}\delta_{pq}\delta(t - t') \tag{14}$$

$$\langle\boldsymbol{\eta}_i(t)\rangle = \mathbf{0} \tag{15}$$

The third term describes the 'drag' effect of the EVL on the DCL. The drag term is computed by using barycentric coordinates considering the velocity of the nearest 3 EVL cell vertices, where the barycentric coefficient is computed with equation (2). Finally, the fourth term is a noise term with amplitude $\sigma$ which describes the autonomous random movement of DCL cells. The notation $\eta_i^p(t)$ indicates the component $p$ of the noise of the particle $i$ at time $t$.

In addition, a defined temporal progression of DCL cell division was implemented to match the experimental data. In the experiments, DCL cells experience one round of asynchronous cell division during the time frame analysed (Supplementary Fig. 2; Supplementary Movie 9). In simulations, cell divisions were implemented at defined times in order to match the total number of DCL cells with that seen in the experiments (Supplementary Movie 9), with the sole restriction that a given DCL cell can only divide once. To match the reduction of DCL cell volume to one half observed in the experiments after cell divisions, we implemented in the simulations a reduction of the radius of the DCL cell to $(1/2)^{1/3}$ of the initial value after division.

**Target functions and fitting procedures for physical models.** Statistical and geometrical quantities obtained from experiments and simulations were used to construct target functions to perform an optimization process and find the optimal values of free parameters of the models that yielded the best agreement with the experimental movies (Supplementary Table 1). We used the Nelder-Mead or Simplex method[46] to find the local minimum values of the target functions. To secure that local minimum values were global we performed a sampling of the values of the free parameters (for example, Fig. 4k). For EVL cells, we used the comparison of areas of corresponding EVL cells between simulation and experiment as target function to obtain the value of $U_0$ (equation (5); Supplementary Fig. 18a,b). In the case of DCL cells, the target function consisted in the comparison between simulation and experiment of four statistical quantities: the distance between the six nearest neighbouring DCL cells (Supplementary Fig. 18c), the probability density function of the distance of DCL cells to the nearest EVL cell border (Supplementary Fig. 18d), the mean square displacement of DCL cells (Supplementary Fig. 18e) and the probability density function of the displacement of DCL cells between two consecutive frames (Supplementary Fig. 18f). With the optimization of the previous target functions we determined the optimal values for $\sigma$ (in equation (14)), $U_1$ (in equation (12)), $U_2$ (in equation (13)) and $D$ (in equation (9)).

**Optimal parameters of physical models.** Supplementary Table 1 summarizes the parameters used in the models to describe EVL expansion and DCL spreading. Fixed parameters were obtained from experimental data while free parameters (asterisks) that better reproduced the experimental results were obtained after an optimization process using the defined target functions (see above and Supplementary Figs 11 and 18).

**Data availability.** The authors declare that all data supporting the findings of this study are available within the article and its Supplementary Information files or from corresponding author upon reasonable request.

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

## Acknowledgements

This work was supported by the Chilean National Commission for Scientific and Technological Research (FONDECYT 3130598 to G.R.; FONDECYT 3140447 and 11161033 to M.C.; FONDECYT 3140444 to V.C.; FONDECYT 1161274, FONDAP 15150012 and FONDEQUIP EQM130051 to M.L.C.; FONDECYT 1151029 and FONDEQUIP EQM140119 to S.H.; Ring Initiative ACT1402 to M.L.C. and S.H.), the Chilean Millennium Science Initiative (grant P09-015-F to M.L.C. and S.H.), CORFO (16CTTS-66390 to S.H.), and DAAD (57220037 & 57168868 to S.H.). We thank Ana-María Lennon-Duménil and Rodrigo Soto for critical reading of the manuscript, and Carl-Philipp Heisenberg for providing the *N-ter-MYPT1 and rhoA* constructs.

## Author contributions

G.R. performed all the experimental approaches in annual killifish and *Fundulopanchax gardneri*. M.C. develop and applied methodologies for segmentation, tracking and the estimation of parameters of cell shape and position. N.S. conceived, numerically implemented and performed the simulations of the physical models. D.F. and V.C. performed segmentation and quantification of cell surface contact areas. M.T. helped in cloning of the annual killifish *e-cad* and the design of the dn Ecad construct. S.H. participated in the design of quantitative approaches and discussion of results. M.L.C. and G.R. designed the experiments and interpreted the results. M.L.C. supervised the project and wrote the paper.

## Additional information

**Competing interests:** The authors declare no competing financial interests.

