## [Peer Review File · Nature Communications]

Reviewer #1 (Remarks to the Author)

The manuscript NCOMMS-16-14311-T "A novel role for extra-embryonic tissue in driving early embryo morphogenesis in vertebrates" reports a detailed study of the displacement of the deep cell layer and the enveloping epithelium during epiboly of the annual killifish, which provides a novel insight into the role of the enveloping epithelium in the spreading of the deep cells. The experimental part of the study appears to be scientifically valid, and the conclusions reached based on the experiments seem plausible and interesting. The manuscript is very well written, clear and complete, and the body of references is suitable. However, the modeling part of the study is much less convincing. As elaborated below I do have a serious conceptual reservation concerning the value of the model, which provides little insight. I cannot recommend the manuscript for publication.

1. The overall logic of the model is entirely phenomenological and overengineered in the sense that it contains a lot of effects. Experiments suggest that DCL cells are dragged by the epithelium and this effect is built into the model; experiments suggest that the motion of DCL cells is not completely deterministic and this effect is built into the model; and experiments suggest that the DCL cells are attracted to the borders between EVL cell and this too is built into the model. From this perspective, it is not very surprising that the model should reproduce the observations. Why is it then needed? What does one learn from the model?

I would much more appreciate a theoretical explanation devoid one or more of these processes but still able to reproduce the observations. For example, is the drag on DCL cell by EVL cell borders not enough? Does one really need the overall non-distributed drag force? Fig. 4g suggests that this is not the case but with a different border potential it could happen. If diffusion were fast enough then the DCL cells could well explore all of the available projected area (which increases because EVL spreads). No explicit drag would be needed in this case. (I do know that the diffusion coefficient was determined experimentally, but the experimentally observed diffusion is probably not a free random walk because of the interaction of DCL cells with each other and with EVL cells. If so, is it OK to include these interactions again in the model explicitly as forces?)

It is possible that a more insightful theory does not exist. Still, the authors do not seem to have tried.

2. The vertex-based model itself is rather detached from real cells. While one may think of spring-like interactions between neighboring vertices, there probably are no such springs between non-neighboring vertices of a given cell. Secondly, it seems to me that in the model, each cell is described by Hookean springs connecting all of its neighboring vertices and all "opposites neighboring vertices" - are the latter all the diagonals? Either way, this seems like too many springs to me. Moreover, their initial lengths are, if I understand it correctly, defined by vertex positions measured at $t = 0$. If so, it appears to me that the model contains many parameters (although Supplementary Table 1 lists only 4 as free parameters), and is thus almost bound to reproduce the experiments.

At the same time, some phenomena that may be of relevance are not included. E.g., the friction of a vertex of EVL may well depend on the number of DCL cells dragged by the vertex. More importantly, I am not certain that the friction coefficients of the EVL and DCL cells (which are not explicitly present in the equations of motion Eqs. (3) and (9)) need to be the same. Sure enough, one of them can be used to scale the force [say in Eq. (3)] but the other one then remains. It seems to me that the left-hand side of Eq. (9) should have a dimensionless factor parametrizing the relative friction coefficients of the EVL and DCL cells - the correspondence between the displacements of the two is fixed by introducing the velocity of the EVL cells $\langle v^{EVL} \rangle$ on the right-hand side of Eq. (9).

3. I am not certain that the quantities used to judge the performance of the model (Fig. 4c-f and

g-j) are necessarily the best choice. For example, I imagine that choosing the distance to 3 nearest neighbors was motivated by the observation that the cells go to the borders of EVL cells but perhaps looking at the distance to the nearest 2 neighbors may be better. Or to the nearest 6? What exactly is the "probability distribution of the distance of DCI cells to EVL border cells" (p. 29) shown in Fig. 4e? Do the authors plot the width of this distribution vs. time? The distribution itself is not plotted in this figure.

I definitely miss arguments justifying the choice of these quantities as figures of merit.

4. Comments concerning presentation:

a. Are hexagons in Fig. 1b representing EVL cells drawn to scale or just schematic? What about the circles? What do the insets in Fig. 1c show? Are they really needed?

b. What is r_j in Eq. (13)? The center of (the nearest?) EVL cell or one of its vertices? What is DCL cell volume (p. 19)? Is this the projected cell area? How is this computed in the model?

c. Fig. 4l: "strenght" -> "strength"

d. "R= 592" in p. 15 is probably "R=592 μ m".

Reviewer #2 (Remarks to the Author)

Report on:

"A novel role for extra-embryonic tissue in driving early embryo morphogenesis in vertebrates" submitted for publication in Nature Communications.

I read this manuscript with much interest and pleasure. I will certainly recommend publication after many corrections are addressed, see below. I believe they can be done within a reasonable time and would significantly improve the clarity of the manuscript. The physics / mechanics part is sometimes weak, and authors should down tone some claims to adapt them to the actual results. I will be happy to review the revised version.

Numbers indicate pages and lines.

p1 L13

«mathematical»  replace with either «physical», «computer» or «numerical»

p1 L17

«adhesive and tensile signals arising from the expansion of extra-embryonic tissue»

- clarify, for instance:

«signals arising from the expansion of extra-embryonic tissue, mediated by cell membrane adhesion and tension»

p2 L20

quote and discuss also:

Behrndt Science 338, 257 (2012);

Xiong Cell 159, 415, 2014

Campinho Nature Cell Biology 15, 1405 (2013)

p3 L8

quote Supp Fig 3 here too

p3 L12

«is tightly coupled to»

-
«parallels closely»

p 3 L13
«suggests that»

-
«raises the questions whether»

p 3 L14
«that»
-
«whether»

p 5 L3
«found»
-
«observed»

p5 L7
«finding was revealed»
-
«was suggested»

Note: a proof would require a direct measurement of cortical tension, by mechanical means

p5L12
«deformability»
-
«deformation»

Note: the observation shows a deformation ; deformability would require to apply and quantify a perturbation, and quantify the corresponding deformation

p5 L21-23:
Fig 3k and movies 7, 8 could be better quantified to extract more information. Even a few manual measurements should suffice.
Statistical tests would indicate that differences are significant.

p5 L25: summarize the three findings (without overclaiming)

p5 L30
«number»
-
«density»

p6 L5
«recruitment»
-
«density»

p6 L6-8
clarify the possible mechanism by which a tension relaxation could be transmitted?

p6 L14
«short-range attraction»

-

«contact interaction»

p6 L15

«tissue spreading»

-

«DCL migration and clustering along EVL cell borders»

p6L16: see p1 L13

p7 L29-30

clarify the possible mechanical signals which could exert dragging and short-range attraction forces?

p8 L19

Note: future works should include more direct mechanical perturbations and measurements. Eg for Fig 2f, Supp. Movie 8.

p13 L27

provide more details on the volume segmentation procedure, with imaged examples; indicate approximate values of the errors and/or noise, as well as their consequences on the results presented in the paper

p16 L16

«17» is in exponent

p16 L20: remove brackets

p17 L3

I understand that a parabolic potential is necessary to regularize simulations, but since experiments do find an absence of such potential, this choice and its consequences should be explicitly discussed.

p17 L5

It is very important to explain the fitting procedure which yields the values of the free parameters. It should be summarized here.

p17 L15

In vertex models, the cell overexpansion is usually prevented with a target area term. Please explain the motivations and consequences of the present alternative choice.

p18 L9

Eqs 9-15 cannot be read on the file I received.

p19 L3

Are all cells dividing at the same time or at different times?

p19 L5

Since there is no target volume, how is it implemented? By enforcing a decrease in length?

p19 L8-27

As noted above, it is very important to explain the fitting procedure which yields the values of the free parameters. It should be detailed here (the current version provides virtually no explanation).

Fig 2e should be analyzed quantitatively.

Fig 3a caption:
after «followed by the cell» add «barycenter»

Fig 3b panel: remove the "0 transition" to make the figure legible

Fig 3j caption:
«Quantification of the radius (r) of the sphere that fits into»
-
«Quantification of the curvature radius (r) of»

Fig 3k caption:
«Movie 9»
-
«Movie 8»

Fig 3l caption:
«miss-shaped»
-
«misshaped»

Note : How is «misshaped» defined here? concavity of some edges, existence of an S-shaped edge, four-fold vertex, elongation?

Fig 4a caption:
«participles»
-
«particles»

Fig 4c-f caption:
«dynamic functions of DCL»
-
«functions of DCL»

Fig 4f panel:
«Simulation»
-
«Complete sim.»

Fig 4k and 4l: exchange them, so that strength of EVL is defined when it is used first;
caption: mention that the greyscale bar codes for strength of EVL;
caption: after «Predictions of» add «effect of»
panel: correct typo «Strenght»  «Strength».

Fig 4m panel:
«Simulation»
-
«Complete sim.»

Fig. 4m-r caption:
add a general caption for m-r, then a specific caption for m and n

Fig 4 m-s panels:
move slightly right the letters m-s so that they fall above their corresponding panels

Fig 4r,s caption:

«Data from panels r and s were extracted from Supplementary Movie 7.»

- please correct

Fig. 4s panel:

Provide quantification of 6 conditions m-r, represented as in Fig. 4i

Fig. S3a, b panels:

Replace 2 panels of snapshots at 40 and 60 percent epiboly with three panels of snapshots at 0, 40 and 60 percent epiboly, noted panels a-c.

Then keep the two bottom panels, which are in the intervals 0-40 and 40-60, and note them d, e.

At least here (and if possible throughout the article), make visible the DCLs which are far from the EVL boundary. So replace the graded red scale with a scale from red (border) to blue (center), or even from blue (border) to red (center).

Fig. S6b panel:

Provide a quantitative measurement to show that EVL movement remains unaffected.

Fig. S6d,e caption:

Define «elongation» and «flatness», explain how they are measured in practice

Fig. S6f caption:

Replace «entropy» with «entropy index», define it, explain how it is measured in practice.

Fig. S8d-f caption:

is this region the square indicated in a,b? If yes, mention it.

Is it a different image, a different experiment?

Fig. S8g-i caption:

where is this image taken?

Fig. S9b panel:

indicate the scale

Fig. S9c caption:

«A gradual increase of E-cad signal is observed within the first 30µm from the EVL cell border, which is followed by a sharp increase at the cell border.»

-

Either suppress «which is followed by a sharp increase at the cell border», or add two arrows on the graph and explain where are the three regions on the graph.

Fig. S9f caption:

«is lost»: show whether this claim is significant

Fig. S9, caption, last sentence:

Move this sentence to the main text. Mention explicitly that you cannot exclude there exists a gradient of another protein.

Fig. S11b panel:

I would plot the total number of DCL rather than their probability; and compare it with the random case for a cell of same size, rather than that of a 120 µm cell.

Fig. S11c caption:

«Summary«

«Schematics»

Fig. S11d caption:

«miss-shaped»

-

«misshaped»

Fig. S11e caption:

is the distance measured inside the EVL cell, outside, or both?

Fig. S12a-g panels:

lowercase letters

Fig. S13 caption:

«supper» - «super»

Fig. S14g-i caption:

is the transition from «close to random» to «preferential distribution» significant? Is there a statistical test?

Table:

Remove «from experiments».

Remove asterisks from symbols of free parameters.

Replace «DC» with «DCL».

Remove the first column of unit dimensions, since they are explicit units on the right (and since the two first lines are incorrect).

Remove square brackets around units.

Remove brackets around parameter names.

Add a separate column containing an asterisk if and only if the parameter is free.

Provide a title to all columns or to none.

Supp. Movie 7:

Add arrows on the first image (or provide a still image with arrows) to indicate the cells.

Supp. Movie 9:

Color cells differently at a division (or: after a division).

Reviewer #3 (Remarks to the Author)

This manuscript uses a very interesting non-model vertebrate system (killifish) to investigate mechanisms of in vivo cell spreading by means of cell-cell interactions. The authors show that the deep cell layer (DCL) interacts with cells of the enveloping layer (EVL) to facilitate cell spreading during the process of epiboly. The interactions involve E-cadherin mediated cell-cell adhesion as well as tension within the EVL cells. Overall the data is well presented and the results advance our understanding of how cells can move in an in vivo context. I do have a few questions and concerns to be addressed.

It seems, although it is not clear, that many of the analyses are done with cells from a single embryo. For the sake of reproducibility, it would be useful to include the total number of embryos analyzed and the consistency of the results between embryos, particularly in cases where experimental manipulations are done.

For the e-cadherin MO experiments in figure 2, it appears (although not mentioned in the figure legend) that the analysis was done at approximately 48hpf (blastocyst stage). If this is the case,

the splice blocking MO will not be inhibiting the maternal spliced transcript (as shown in the supplemental figure) and thus it is unclear if this is a specific phenotype. The dominant negative E-cad nicely phenocopies the MO, so some clarification of the stage (hpf) and whether the MO would be expected to be working is warranted.

In the introduction, page 2 lines 11, 14, 15, the authors state that the spreading of the DCL takes place in an environment that is devoid of patterning signals and that lacks an organized ECM. It is unclear how the authors come to these conclusions as the papers and supplemental figure cited do not address this. Although the data clearly suggest a cell-cell interaction for DCL spreading, these statements imply that this is the only mechanism by which they spread without providing the evidence.

On page 4 line 21, the figure reference should be figure 3c, not 2c.

On page 4 line 29, the text states that cells use local cues to direct their autonomous random motility. It seems that if it is directed motility it cannot also be random – I think what is meant here is that they switch from random to directed migration when they approach the border?

I think it would be helpful to move some of the supplemental figures to the main text (particularly supplemental figure 11) since the manuscript is currently well under the figure/table limit.

Response to Reviewers:

We thank all reviewers for their constructive criticisms that we believe have allowed us to increase the clarity and quality of our manuscript.

Reviewer #1 (Remarks to the Author):

The manuscript NCOMMS-16-14311-T "A novel role for extra-embryonic tissue in driving early embryo morphogenesis in vertebrates" reports a detailed study of the displacement of the deep cell layer and the enveloping epithelium during epiboly of the annual killifish, which provides a novel insight into the role of the enveloping epithelium in the spreading of the deep cells. The experimental part of the study appears to be scientifically valid, and the conclusions reached based on the experiments seem plausible and interesting. The manuscript is very well written, clear and complete, and the body of references is suitable. However, the modeling part of the study is much less convincing. As elaborated below I do have a serious conceptual reservation concerning the value of the model, which provides little insight. I cannot recommend the manuscript for publication.

Q1. The overall logic of the model is entirely phenomenological and overengineered in the sense that it contains a lot of effects. Experiments suggest that DCL cells are dragged by the epithelium and this effect is built into the model; experiments suggest that the motion of DCL cells is not completely deterministic and this effect is built into the model; and experiments suggest that the DCL cells are attracted to the borders between EVL cell and this too is built into the model. From this perspective, it is not very surprising that the model should reproduce the observations. Why is it then needed? What does one learn from the model?

Response: The model we propose is indeed phenomenological in the sense of the included elements. However, it contributes: (i) to verify if the inclusion of EVL drag of the DCL, non-determinist DCL movement, and DCL attraction to EVL cells border is sufficient to reproduce the experimental results; and (ii) to show that the principal determinant of vegetal-ward directed DCL spreading is the non-autonomous dragging by the EVL, over the multiple elements included.

(i) From the experiments we see that spreading of the embryonic DCL involves a combination of autonomous random motility, a dragging effect by the expanding extra-embryonic surface epithelium (the EVL) and short-range attraction by EVL cell borders. With the model we wanted to test if the interaction of these three components was sufficient to drive tissue spreading of the DCL cells. To this aim we constructed a mathematical model based on interacting point particles confined to move on the surface of a sphere. The numerical simulations showed us that the dynamics and spatial configuration of both EVL surface expansion and DCL spreading observed in the experiments is well reproduced, supporting the sufficiency of the model.

(ii) The mathematical model also allowed us to establish the individual contribution of autonomous and non-autonomous forces to DCL spreading. We show in Fig. 4h-k a detailed description of the results of the model when we consider each force or a

combination of them. We see from this comparison that the most important force is the dragging effect of EVL cells on the DCL cells. Also, it seems that the EVL border potential does not play a relevant role in the dispersion of the DCL cells.

Q2. I would much more appreciate a theoretical explanation devoid one or more of these processes but still able to reproduce the observations. For example, is the drag on DCL cell by EVL cell borders not enough? Does one really need the overall non-distributed drag force? Fig. 4g suggests that this is not the case but with a different border potential it could happen.

Response: To examine the possibility mentioned by the reviewer that DCL attraction by EVL border potential alone could explain DCL spreading (or noise amplitude, or potential intensity, or shape) we now explored separately each component. The main result is that taking into account only EVL cell border attraction potential does not reproduce spreading; specifically frame-to-frame displacement shows a poor agreement. The same result is observed with the parabolic or a Gaussian potential (finally used in the paper simulations). We further explore the approach by taking pairs of elements. The details are explained below and in the new Supplementary Fig. 13.

The figures of merit that we use to analyse the sufficiency of our model were: the probability density function of displacement of the DCL between frame, the probability density function of the distance between DCL and the nearest EVL edge, the root mean square displacement for DCL cells, and the distance between the 6 neighbouring DCL cells.

To test the sufficiency of each of the forces used in the model, we ran simulations where we considered each force at a time: noise amplitude, EVL drag over DCL, and EVL cell border attraction potential. With this we analysed if only with one of the ingredients we could reproduce the experimental data. From Supplementary Fig. 13, we see that it is not possible to have an agreement between experiment and simulation in all figures of merit taking a single ingredient. Among these 3 ingredients, the best agreement between simulation and experiment occurs when only drag is considered, but the agreement of the probability density function for the displacement between frames is very bad. From the other two figures mentioned above we see that considering only the EVL cell border attraction potential or only the noise delivers a rather poor agreement when comparing experiment and model.

Then, we ran simulations where we took all possible combinations of two ingredients. As shown in Supplementary Fig. 13, agreement with the experiment is better than in the case of considering only one ingredient but still some figures of merit are not in agreement with experiment depending on the combination of variables considered. We see that taking noise and drag the agreement is quite good between the model and experiment; only the probability density of the DCL distance to the nearest EVL cell border is badly reproduced by the model. This indicates that in the experiment there is an attraction of the DCL to the edges of the EVL. Adding the attraction potential towards the EVL cell borders we see that we have an agreement in the four figures of merit that we consider.

Q3. If diffusion were fast enough then the DCL cells could well explore all of the available projected area (which increases because EVL spreads). No explicit drag would be needed in this case.

Response: Yes, this is true but with a higher noise amplitude the probability density function of displacements ceases to be in agreement with the experiment, see answer above.

Q4. I do know that the diffusion coefficient was determined experimentally, but the experimentally observed diffusion is probably not a free random walk because of the interaction of DCL cells with each other and with EVL cells. If so, is it OK to include these interactions again in the model explicitly as forces?

Response: We agree with this observation, therefore, now we consider the amplitude of noise as an adjustable parameter.

It is possible that a more insightful theory does not exist. Still, the authors do not seem to have tried.

Q5. The vertex-based model itself is rather detached from real cells. While one may think of spring-like interactions between neighboring vertices, there probably are no such springs between non-neighboring vertices of a given cell. Secondly, it seems to me that in the model, each cell is described by Hookean springs connecting all of its neighboring vertices and all "opposites neighboring vertices" - are the latter all the diagonals? Either way, this seems like too many springs to me. Moreover, their initial lengths are, if I understand it correctly, defined by vertex positions measured at $t = 0$. If so, it appears to me that the model contains many parameters (although Supplementary Table 1 lists only 4 as free parameters), and is thus almost bound to reproduce the experiments.

Response: In the EVL model, the springs at the edges of the EVL cell are considered to represent the actin cortical band observed in the experiments. On the other hand, the diagonal springs models the behaviour of the cytoskeleton of the EVL. With respect to the length of the springs at $t = 0$ this corresponds to the reference configuration of the epithelium obtained from the experiment at the beginning of the observation, so it is not a free parameter of the model, and thus it is not included in Table 1.

Q6. At the same time, some phenomena that may be of relevance are not included. E.g., the friction of a vertex of EVL may well depend on the number of DCL cells dragged by the vertex.

Response: One of the hypotheses of our model, to simplify the description, is that the DCL are primarily influenced by the EVL, and not the contrary. Although we can not currently rule out that the DCL has some influence on EVL movement, the fact that epiboly of the EVL is not affected by abrogation of E-cad function while the DCL is severely affected, argues at least in favour of a stronger influence of the EVL on the DCL than the opposite possibility.

Q7. More importantly, I am not certain that the friction coefficients of the EVL and DCL cells (which are not explicitly present in the equations of motion Eqs. (3) and (9)] need to be the same. Sure enough, one of them can be used to scale the force [say in Eq. (3)] but the other one then remains. It seems to me that the left-hand side of Eq. (9) should have a dimensionless factor parametrizing the relative friction coefficients of the EVL and DCL cells - the correspondence between the displacements of the two is fixed by introducing the velocity of the EVL cells $\langle v^{\text{EVL}} \rangle$ on the right-hand side of Eq. (9).

Response: The friction coefficients of equations (3) and (9) were absorbed by the right hand side parameters in equations (3) and (9), thus the forces have units of velocity.

Q8. I am not certain that the quantities used to judge the performance of the model (Fig. 4c-f and g-j) are necessarily the best choice. For example, I imagine that choosing the distance to 3 nearest neighbors was motivated by the observation that the cells go to the borders of EVL cells but perhaps looking at the distance to the nearest 2 neighbors may be better. Or to the nearest 6? What exactly is the "probability distribution of the distance of DCL cells to EVL border cells" (p. 29) shown in Fig. 4e? Do the authors plot the width of this distribution vs. time? The distribution itself is not plotted in this figure.

I definitely miss arguments justifying the choice of these quantities as figures of merit.

Response: We cannot rule out that other quantities can also be used to judge the performance of the model, but as suggested we change the distance from 3 to 6 neighbours and included more temporal aspects using the mean square displacement (MSD). The figures of merit that we now use to analyse the sufficiency of our model are: (i) the probability density of the distance between DCL and the nearest EVL edge, (ii) the distance between the 6 neighbouring DCL cells, (iii) the probability density of displacement of the DCL between frame, and (iv) the MSD for DCL.

The choice of the figures of merits was to give numerical values to the spatial and temporal characteristics of the dynamics of the DCL cells. Figures of merits (i) and (ii) quantify the spatial distribution of the DCL cells on the sphere at each time instant. On the other hand, the figures of merits (iii) and (iv) measure DCL cells temporal dynamics on the sphere by taking pairs of consecutive times (iii), or dynamics over both short and long terms dynamics in the case of MSD (iv). Supplementary Figure 13 shows in detail the dependence of (i)-(iv) on the model parameters.

About the probability density distribution of the distance from DCL to the nearest EVL cell border is the probability per unit length that a DCL is at a given distance from an edge EVL, this probability density is now in Fig. 4. This quantity was also computed for each frame as suggested, however, due to the few data points for a given frame, the distribution is noisier than when computed for the full set of frames.

4. Comments concerning presentation:

a. Are hexagons in Fig. 1b representing EVL cells drawn to scale or just schematic?

What about the circles?

What do the insets in Fig. 1c show? Are they really needed?

Response: In Fig. 1b, the blue polygons and red circles above the curves depict EVL and DCL cells drawn to scale, respectively. In Fig. 1c, the insets indicate how the total area covered by the EVL and the convex area covered by the underlying DCL cells are measured. This is now indicated in the caption of Fig. 1.

b. What is r_j in Eq. (13)?

The center of (the nearest?) EVL cell or one of its vertices?

Response: $|r_i - r_j|$ is the orthogonal distance between DCL and EVL cell border

What is DCL cell volume (p. 19)? Is this the projected cell area? How is this computed in the model?

Response: yes, volume correspond to the projected cell area, which is determined by the radius. The key parameter used for the interactions between DCL cells in the simulations is the radius.

c. Fig. 4l: "strenght" -> "strength"

Response: done.

d. "R= 592" in p. 15 is probably "R=592 μ m".

Response: yes, it has been corrected.

Reviewer #2 (Remarks to the Author):

I read this manuscript with much interest and pleasure. I will certainly recommend publication after many corrections are addressed, see below. I believe they can be done within a reasonable time and would significantly improve the clarity of the manuscript. The physics / mechanics part is sometimes weak, and authors should down tone some claims to adapt them to the actual results. I will be happy to review the revised version.

Numbers indicate pages and lines.

p1 L13

«mathematical»  replace with either «physical», «computer» or «numerical»

Response: “mathematical” replaced by “physical” in p1L13 and elsewhere.

p1 L17

«adhesive and tensile signals arising from the expansion of extra-embryonic tissue»

- clarify, for instance:

«signals arising from the expansion of extra-embryonic tissue, mediated by cell membrane adhesion and tension»

Response: changed to “... couple their autonomous random motility to non-autonomous signals arising from the expansion of extra-embryonic tissue, which is used by embryonic cells as a substrate for migration, in a process mediated by cell membrane adhesion and tension.”

p2 L20

quote and discuss also:

Behrndt Science 338, 257 (2012);

Xiong Cell 159, 415, 2014

Campinho Nature Cell Biology 15, 1405 (2013)

Response: we have incorporated the Behrndt and Campinho references in the context of the following sentence: “The EVL, which lies above the DCL, spreads by an actomyosin dependent mechanism generated at the circumferential margin that generates pulling forces and tension anisotropy within the epithelium (Behrndt, Salbreux et al. 2012, Campinho, Behrndt et al. 2013).” The work by Xiong et al about has not been included in the section (but it has in the discussion) as it refers to a stage of EVL morphogenesis prior to epiboly.

p3 L8

quote Supp Fig 3 here too

Response: done.

p3 L12

«is tightly coupled to»

-

«parallels closely»

Response: done.

p 3 L13

«suggests that»

-

«raises the questions whether»

Response: done.

p 3 L14

«that»

-

«whether»

Response: done.

p 5 L3

«found»

-

«observed»

Response: done

p5 L7

«finding was revealed»

-

«was suggested»

Note: a proof would require a direct measurement of cortical tension, by mechanical means

Response: done

p5L12

«deformability»

-

«deformation»

Note: the observation shows a deformation; deformability would require to apply and quantify a perturbation, and quantify the corresponding deformation

Response: done

p5 L21-23:

Fig 3k and movies 7, 8 could be better quantified to extract more information. Even a few manual measurements should suffice.

Statistical tests would indicate that differences are significant.

Response: a graph showing the quantification of DCL cell eccentricity during EVL cell fusion has been incorporated in the Supplementary Fig. 11.

p5 L25: summarize the three findings (without overclaiming)

Response: we have rephrased the findings described in this paragraph as: “Altogether, these findings suggest that cortical tension/stiffness is enhanced at EVL cell borders

and that this local property of the cellular substrate favours adhesive and tensile cell-substrate interactions that foster the migration of DCL cells”.

p5 L30

«number»

-

«density»

Response: done.

p6 L5

«recruitment»

-

«density»

Response: done.

p6 L6-8

clarify the possible mechanism by which a tension relaxation could be transmitted?

Response: we have added the following sentence: “How local relaxation in tension was transmitted across the epithelium is yet to be determined but as cell number can not be modified in the EVL due to failed cytokinesis, tissue relaxation possibly involve changes in cell shape, volume and/or reorganisation as seen in other epithelia under geometrical constraints (Behrndt, Salbreux et al. 2012, Campinho, Behrndt et al. 2013, Xiong, Ma et al. 2014).”

p6 L14

«short-range attraction»

-

«contact interaction»

Response: done.

p6 L15

«tissue spreading»

-

«DCL migration and clustering along EVL cell borders»

Response: done.

p6L16: see p1 L13

Response: done.

p7 L29-30

clarify the possible mechanical signals which could exert dragging and short-range attraction forces?

Response: we have clarified this point by adding the following paragraph: “Tissue-tissue mechanical coupling requires E-cad dependent cell-cell adhesive and tensile interactions that result in both dragging of embryonic cells by the vegetal movement of the EVL and short range attraction towards regions of increased cortical tension, the

EVL cell borders. Together, autonomous motility, dragging and short range attraction forces direct cell migration and the spreading of embryonic tissue following structural features of the extra-embryonic cellular substrate in a way reminiscent to contact guidance (Weiss 1961, Carter 1965).”.

p8 L19

Note: future works should include more direct mechanical perturbations and measurements. Eg for Fig 2f, Supp. Movie 8.

Response: done.

p13 L27

provide more details on the volume segmentation procedure, with imaged examples; indicate approximate values of the errors and/or noise, as well as their consequences on the results presented in the paper

Response: we have added the new Supplementary Fig. 16 to address this issue. In this figure we show that the error of measuring DCL distances on the deformed surface relative to real z-axis ranges from around 0% (i.e. for two DCL cells in contact) to 3% (i.e. for very distant DCL cells). We thus consider that this amount of error has no major effects in the quantifications and simulations.

p16 L16

«17» is in exponent

Response: done.

p16 L20: remove brackets

Response: done.

p17 L3

I understand that a parabolic potential is necessary to regularize simulations, but since experiments do find an absence of such potential, this choice and its consequences should be explicitly discussed.

Response: to simulate EVL cell border attraction, we tested gaussian and parabolic potentials (new Supplementary Fig. 19). Gaussian and parabolic potentials give similar results for most figures of merit. However, the probability distribution of the distance of DCL cells to EVL cell borders obtained after a gaussian potential fits better the smooth increase towards shorter distances seen in the experiments, compared to the parabolic potential (Supplementary Fig. 19c). This behaviour is also reminiscent of the smooth transition in EVL cell surface deformation (Fig. 3j) and shared surface with DCL cells (Fig. 3h) observed as a DCL approaches the EVL cell border. For all these reasons, we considered that a gaussian potential was the best choice for simulations.

p17 L5

It is very important to explain the fitting procedure which yields the values of the free parameters. It should be summarized here.

Response: an explanation of the fitting procedure has been included in the section “Target functions and fitting procedures for EVL and DCL physical models.” (Methods, section physical modelling).

p17 L15

In vertex models, the cell overexpansion is usually prevented with a target area term. Please explain the motivations and consequences of the present alternative choice.

Response: we found that for the time interval addressed in this paper (from the beginning of epiboly until the EVL margin is around the equator of the embryo) our spring model fits well the experimental data and EVL cell overexpansion seems not to be a relevant factor. Indeed, we observe no appreciable differences in the area of EVL cells when comparing simulations using our model and the vertex model (see Reviewer Fig. 1 below). Only at later stages of epiboly, when the EVL margin approaches the vegetal pole we start to observe EVL overexpansion in our model, which seems prevented in the vortex model (data not shown). As our model of springs recapitulated the EVL experimental data in the time-frame of analysis used in this paper, and because the EVL model was implemented primarily to provide a substrate context for the analysis of DCL-EVL interactions (and not to explain the forces driving the expansion of the EVL per se), we decided to keep this model regardless of the limitations observed at later stages of epiboly. In a future work, we will compare the pros and cons of our model and the vortex model in the context of analysing epithelial expansion.

Figure Reviewer 1: The figure shows a comparison of the temporal changes in surface area of 12 individual EVL cells using the spring model of this paper (black dashed line) and the vertex model (green line).

p18 L9

Eqs 9-15 cannot be read on the file I received.

Response: corrected.

p19 L3

Are all cells dividing at the same time or at different times?

Response: in the simulations, the temporal pattern of DCL cell division mimics the temporal progression of cell division observed in the experiments, which is asynchronous. This was achieved by comparing at each time the total number of DCL cells between the simulation and experiments, and implement events of division in the simulation such as to equal the total number of DCL cells between the two conditions. The only restriction, which was taken from our experimental observations, was that a single DCL cell can only divide once in the period of analysis.

p19 L5

Since there is no target volume, how is it implemented? By enforcing a decrease in length?

Response: To match the reduction of DCL cell volume to one half observed in the experiments after cell division, we implemented in the simulations a reduction of the radius to $(1/2)^{1/3}$ of the initial value in daughter cells after events of cell division.

p19 L8-27

As noted above, it is very important to explain the fitting procedure which yields the values of the free parameters. It should be detailed here (the current version provides virtually no explanation).

Response: as stated above, an explanation of the fitting procedure has been included in the section "Target functions and fitting procedures for EVL and DCL physical models." (Methods, section physical modelling).

Fig 2e should be analyzed quantitatively.

Response: a quantification of the effect of dn-Ecad has been incorporated in panels e to g of Fig. 2.

Fig 3a caption:

after «followed by the cell» add «barycenter»

Response: done.

Fig 3b panel: remove the "0 transition" to make the figure legible

Response: done.

Fig 3j caption:

«Quantification of the radius (r) of the sphere that fits into»

-

«Quantification of the curvature radius (r) of»

Response: done.

Fig 3k caption:

«Movie 9»

-

«Movie 8»

Response: done.

Fig 3l caption:

«miss-shaped»

-

«misshaped»

Note : How is «misshaped» defined here? concavity of some edges, existence of an S-shaped edge, four-fold vertex, elongation?

Response: we have changed “misshaped” by “more concave”.

Fig 4a caption:

«participles»

-

«particles»

Response: done.

Fig 4c-f caption:

«dynamic functions of DCL»

-

«functions of DCL»

Response: done.

Fig 4f panel:

«Simulation»

-

«Complete sim.»

Response: done.

Fig 4k and 4l: exchange them, so that strength of EVL is defined when it is used first;

Response: done.

caption: mention that the greyscale bar codes for strength of EVL;

Response: done.

caption: after «Predictions of» add «effect of»

Response: done.

panel: correct typo «Strenght»  «Strength».

Response: done.

Fig 4m panel:

«Simulation»

-

«Complete sim.»

Response: done.

Fig. 4m-r caption:

add a general caption for m-r, then a specific caption for m and n

Response: done.

Fig 4 m-s panels:

move slightly right the letters m-s so that they fall above their corresponding panels

Response: done.

Fig 4r,s caption:

«Data from panels r and s were extracted from Supplementary Movie 7.»

- please correct

Response: done.

Fig. 4s panel:

Provide quantification of 6 conditions m-r, represented as in Fig. 4i

Response: a quantification of the conditions WT, Rac1-T17N and e-cad-MO have been incorporated in a single graph in panel t of Fig. 4.

Fig. S3a, b panels:

Replace 2 panels of snapshots at 40 and 60 percent epiboly with three panels of snapshots at 0, 40 and 60 percent epiboly, noted panels a-c.

Then keep the two bottom panels, which are in the intervals 0-40 and 40-60, and note them d, e.

Response: done.

At least here (and if possible throughout the article), make visible the DCLs which are far from the EVL boundary. So replace the graded red scale with a scale from red (border) to blue (center), or even from blue (border) to red (center).

Response: done.

Fig. S6b panel:

Provide a quantitative measurement to show that EVL movement remains unaffected.

Response: Quantification of EVL movement in WT and dn E-cad embryos has been incorporated in panel h of Fig. 2.

Fig. S6d,e caption:

Define «elongation» and «flatness», explain how they are measured in practice

Response: done.

Fig. S6f caption:

Replace «entropy» with «entropy index», define it, explain how it is measured in practice.

Response: done. See caption S7d,e.

Fig. S8d-f caption:

is this region the square indicated in a,b? If yes, mention it.

Is it a different image, a different experiment?

Response: yes, d-f are high magnification views of the square regions depicted in a-b. Now indicated in S9 caption.

Fig. S8g-i caption:

where is this image taken?

Response: images in g-i are from a different embryo than those in a-f. It is now indicated in S9 caption.

Fig. S9b panel:

indicate the scale

Response: done.

Fig. S9c caption:

«A gradual increase of E-cad signal is observed within the first 30µm from the EVL cell border, which is followed by a sharp increase at the cell border.»

-

Either suppress «which is followed by a sharp increase at the cell border», or add two arrows on the graph and explain where are the three regions on the graph.

Response: done.

Fig. S9f caption:

«is lost»: show whether this claim is significant

Response: done.

Fig. S9, caption, last sentence:

Move this sentence to the main text. Mention explicitly that you cannot exclude there exists a gradient of another protein.

Response: done.

Fig. S11b panel:

I would plot the total number of DCL rather than their probability; and compare it with the random case for a cell of same size, rather than that of a 120 µm cell.

Response: as suggested, we have included the comparison of the random case in each condition using the corresponding EVL cell size. We decided, however, to keep the plot of probability rather than to change it to cell number to better compare the experimental data with the hypothetical random condition. Also, we have moved the panels of this supplementary figure to Fig. 3 (panels n and o).

Fig. S11c caption:

«Summary«

«Schematics»

Response: done.

Fig. S11d caption:

«miss-shaped»

-

«misshaped»

Response: done.

Fig. S11e caption:

is the distance measured inside the EVL cell, outside, or both?

Response: the distance was measured for each DCL cell, from its centre of mass (defined in 2D on the z-max projection) to the closest EVL cell border. No DCL cells were excluded.

Fig. S12a-g panels:

lowercase letters

Response: done.

Fig. S13 caption:

«supper» - «super»

Response: done.

Fig. S14g-i caption:

is the transition from «close to random» to «preferential distribution» significant? Is there a statistical test?

Response: yes, we found that the probability distribution of DCL distance to EVL cell border was significantly different at 50% epiboly (panel i of new Fig. 15) compared to the distributions observed at the onset of epiboly (panel g) and 30% epiboly (panel h) (Kolmogorov-Smirnov, $p < 0.01$; included in caption).

Table:

Remove «from experiments».

Remove asterisks from symbols of free parameters.

Replace «DC» with «DCL».

Remove the first column of unit dimensions, since they are explicit units on the right (and since the two first lines are incorrect).

Remove square brackets around units.

Remove brackets around parameter names.

Add a separate column containing an asterisk if and only if the parameter is free.

Provide a title to all columns or to none.

Response: done.

Supp. Movie 7:

Add arrows on the first image (or provide a still image with arrows) to indicate the cells.

Response: done.

Supp. Movie 9:

Color cells differently at a division (or: after a division).

Response: done.

Reviewer #3 (Remarks to the Author):

This manuscript uses a very interesting non-model vertebrate system (killifish) to investigate mechanisms of in vivo cell spreading by means of cell-cell interactions. The authors show that the deep cell layer (DCL) interacts with cells of the enveloping layer (EVL) to facilitate cell spreading during the process of epiboly. The interactions involve E-cadherin mediated cell-cell adhesion as well as tension within the EVL cells. Overall the data is well presented and the results advance our understanding of how cells can move in an in vivo context. I do have a few questions and concerns to be addressed.

It seems, although it is not clear, that many of the analyses are done with cells from a single embryo. For the sake of reproducibility, it would be useful to include the total number of embryos analyzed and the consistency of the results between embryos, particularly in cases where experimental manipulations are done.

Response: numbers of embryos and cells analysed are now included in each figure. As a general rule, we analysed at least 3 embryos per condition, and performed the appropriate statistical validations of significance (see Methods). Only for the experimental cases that involved extensive time-lapse tracking and analysis of dynamic data, we present the data of a single embryo as a main figure and replicate the analysis in an additional embryo and show this as supplementary material (i.e. the dynamic analysis of Fig. 1 has been replicated in an additional embryo and included in Supplementary Fig. 3)

For the e-cadherin MO experiments in figure 2, it appears (although not mentioned in the figure legend) that the analysis was done at approximately 48hpf (blastocyst stage). If this is the case, the splice blocking MO will not be inhibiting the maternal spliced transcript (as shown in the supplemental figure) and thus it is unclear if this is a specific phenotype. The dominant negative E-cad nicely phenocopies the MO, so some clarification of the stage (hpf) and whether the MO would be expected to be working is warranted.

Response: we thank the reviewer for making us aware of this point. Indeed, the image corresponded to a 48hpf embryo, just prior to the onset of epiboly, but we intended to provide an image of an epiboly stage. We now include an image of a 40% epiboly embryo (60 hpf), and all quantifications shown in “d” correspond to this stage.

In the introduction, page 2 lines 11, 14, 15, the authors state that the spreading of the DCL takes place in an environment that is devoid of patterning signals and that lacks an organized ECM. It is unclear how the authors come to these conclusions as the papers and supplemental figure cited do not address this. Although the data clearly suggest a cell-cell interaction for DCL spreading, these statements imply that this is the only mechanism by which they spread without providing the evidence.

Response: previous reports in embryos of annual killifishes, describing cell behaviour in the DCL during epiboly, dispersion and reaggregation (e.g. Wourms 1972; Carter and Wourms 1991), as well as global analysis of gene expression (Warner and Podrabsky 2015), combined with our own observations of the expression domains of two genes involved in gastrulation (*brachyury* and *gooseoid*; Pereiro et al. submitted

to Developmental Dynamics), have shown that the DCL during epiboly behaves as an undifferentiated group of cells and do not yet show cell behaviours nor express genetic markers characteristic of the process of gastrulation and embryonic axis formation. These events are only seen well after epiboly finishes and the cells disperse over the yolk, when cells congregate to form an aggregate that is the precursor of the embryo proper in annual killifishes.

We agree with the reviewer that the phrase used in page 2 lines 11, 14, 15, was not clear enough and that we did not include all needed references. Thus, we have incorporated the missing references (with the exception of Pereiro et al. which is still under review) and rephrased the statement to: “... we took advantage of unique developmental features of a non-conventional teleost embryo where undifferentiated mesenchymal-like embryonic cells spread as a collective at very low cell density and in a simple cellular environment, well before the onset of gastrulation and embryonic axis formation (Wourms 1972, Carter and Wourms 1991, Wagner and Podrabsky 2015).”

Regarding the “lack of organised ECM”, an ultrastructural study demonstrating this point was already referenced in the previous version, at the beginning of the second paragraph of the main text (Carter and Wourms 1990). The confusion may have raised from the fact that this reference was placed at the end of a long sentence that addressed two different aspects of DCL spreading. Thus, we have now placed this reference under the specific statement of the ECM.

On page 4 line 21, the figure reference should be figure 3c, not 2c.

Response: done.

On page 4 line 29, the text states that cells use local cues to direct their autonomous random motility. It seems that if it is directed motility it cannot also be random – I think what is meant here is that they switch from random to directed migration when they approach the border?

Response: we have rephrased this sentence as “Together, these findings indicate that DCL cells show autonomous random motility when they are far from the EVL cell border and that they switch from random to directional migration when they approach this region. Thus, EVL cell borders provide short-range cues to attract DCL cell migration.”

I think it would be helpful to move some of the supplemental figures to the main text (particularly supplemental figure 11) since the manuscript is currently well under the figure/table limit.

Response: we have moved the panels of the original Supplementary Fig. 11 to Fig 3 (panels n,o).

References

- Behrndt, M., G. Salbreux, P. Campinho, R. Hauschild, F. Oswald, J. Roensch, S. W. Grill and C. P. Heisenberg (2012). "Forces driving epithelial spreading in zebrafish gastrulation." Science **338**(6104): 257-260.
- Campinho, P., M. Behrndt, J. Ranft, T. Risler, N. Minc and C. P. Heisenberg (2013). "Tension-oriented cell divisions limit anisotropic tissue tension in epithelial spreading during zebrafish epiboly." Nat Cell Biol **15**(12): 1405-1414.
- Carter, C. A. and J. P. Wourms (1991). "Cell behavior during early development in the South American annual fishes of the genus *Cynolebias*." J Morphol **210**(3): 247-266.
- Carter, S. B. (1965). "Principles of cell motility: the direction of cell movement and cancer invasion." Nature **208**(5016): 1183-1187.
- Wagner, J. T. and J. E. Podrabsky (2015). "Gene expression patterns that support novel developmental stress buffering in embryos of the annual killifish *Austrofundulus limnaeus*." Evodevo **6**: 2.
- Weiss, P. (1961). "Guiding principles of cell locomotion and cell aggregation." Experimental Cell Research Suppl. **8**: 260-281.
- Wourms, J. P. (1972). "The developmental biology of annual fishes. II. Naturally occurring dispersion and reaggregation of blastomers during the development of annual fish eggs." J Exp Zool **182**(2): 169-200.
- Xiong, F., W. Ma, T. W. Hiscock, K. R. Mosaliganti, A. R. Tentner, K. A. Brakke, N. Rannou, A. Gelas, L. Souhait, I. A. Swinburne, N. D. Obholzer and S. G. Megason (2014). "Interplay of cell shape and division orientation promotes robust morphogenesis of developing epithelia." Cell **159**(2): 415-427.

Reviewer #1 (Remarks to the Author)

In their response and in the revised version of the manuscript, the authors have addressed the questions that I raised in my first report. There my main concern was the rationale behind the model used and in the revised version the authors have presented additional data which show that indeed all three effects contributing to the displacement of DCL are needed. As a whole, the manuscript warrants publication in Nat. Commun. and I suggest that it be accepted.

Reviewer #2 (Remarks to the Author)

I recommend to publish this revised version.

A few typos left :

- p5 L25

«deformability»

-

«deformation»

- p5 L28

«contractibility»

--

«contractility»

- p6 L7

«tension/stiffness»

--

«tension and/or stiffness»

- p6 L15

idem

- p7 L13

«mimmic»

-

«mimick»

- p8 L9

«works»

--

«work»

- p15 L5

«43» in exponent

- p17 L22

after «1.1» remove «u»

- p20 L22

missing symbol

Reviewer #3 (Remarks to the Author)

I have reviewed the revised manuscript. My concerns regarding the first submission have been adequately addressed by the authors and I'm happy to support publication.

RESPONSE TO REVIEWERS' COMMENTS:

Reviewer #1 (Remarks to the Author):

In their response and in the revised version of the manuscript, the authors have addressed the questions that I raised in my first report. There my main concern was the rationale behind the model used and in the revised version the authors have presented additional data which show that indeed all three effects contributing to the displacement of DCL are needed. As a whole, the manuscript warrants publication in Nat. Commun. and I suggest that it be accepted.

--

Reviewer #2 (Remarks to the Author):

I recommend to publish this revised version.

A few typos left :

- p5 L25
«deformability»
-
«deformation»

Answer: Corrected in the text.

- p5 L28
«contractibility»
--
«contractility»

Answer: Corrected in the text.

- p6 L7
«tension/stiffness»
--
«tension and/or stiffness»

Answer: Corrected in the text.

- p6 L15
idem

Answer: Corrected in the text.

- p7 L13
«mimmic»
-
«mimick»

Answer: Corrected in the text.

- p8 L9
«works»
--
«work»

Answer: Corrected in the text.

- p15 L5
«43» in exponent

Answer: Corrected in the text.

- p17 L22
after «1.1» remove «u»

Answer: Corrected in the text.

- p20 L22
missing symbol

Answer: Corrected in the text.

--

Reviewer #3 (Remarks to the Author):

I have reviewed the revised manuscript. My concerns regarding the first submission have been adequately addressed by the authors and I'm happy to support publication.